# DSTA4D: Rethinking Adaptive Spatio-Temporal Decoupling for Dynamic Point Cloud Videos

## Abstract

Understanding 4D point cloud videos is crucial for intelligent agents to perceive the dynamic changes in their external environment. However, due to the inter-frame time inconsistency and spatial disorder inherent in long-sequence point clouds, designing a unified 4D global model faces significant challenges. Existing methods primarily rely on static, monolithic network architectures that apply a uniform computational pipeline to all input data. This approach neglects the differences in spatio-temporal complexity across videos, resulting in inefficient resource allocation and limiting model's performance. To address these issues, we present a novel content-aware 4D point cloud processing approach, termed DSTA4D, which leverages dynamic spatio-temporal decoupling via adaptive modules. We first propose decoupling temporal and spatial features within the embedding layer, which avoids the complexity of full-process long-term modeling. Second, we introduce a innovative lightweight module: Dynamic Spatio-Temporal Adapter (DST-Adapter). This module dynamically generates gating weights based on the global spatio-temporal features of the input sequence and adaptively fuses features from three parallel streams: identity path, spatial enhancement path, and temporal enhancement path. This content-aware mechanism allows the model to intelligently allocate its computational focus to the most critical feature dimensions. Our experiments on mainstream benchmarks MSR-Action3D (**+5.23%** accuracy), NTU RGBD (**+1%** accuracy) and Synthia4D (**+1.36%** mIoU) show significant performance gains, offering a more efficient and intelligent adaptive modeling paradigm for point cloud video understanding.

## 1 Introduction

4D point clouds represent the fusion of three-dimensional space and one-dimensional time. As a rich data modality, it effectively captures both 3D geometric structures and temporal dynamics. Understanding 4D point cloud videos has become a cornerstone of environmental perception in frontier domains such as robotics, autonomous driving, and augmented reality. While substantial progress has been made in static 3D point cloud understanding, designing efficient backbone networks for dynamic 4D point cloud sequences remains a significant challenge. Effectively interpreting the complex dynamics within these videos is crucial for intelligent agents to interact with the physical world. Existing 4D understanding methods typically model the 4D sequences as voxel representations, integrated with convolutional modules(Fan et al., 2022). However, due to the inherent unordered and sparse nature of 4D point cloud data, as well as the inconsistency of points across frames, designing a representation learning model that is low-error, efficient, and robust remains a pressing issue.

Prevailing methods for processing point cloud sequences, whether based on Convolutional Neural Networks (CNNs) (Liu et al., 2019), Transformers(Fan et al., 2021a), or even advanced State Space Models (SSMs) (Liu et al., 2025), predominantly adopt a static and monolithic processing paradigm. This "one-size-fits-all" approach executes an identical computational path for all inputs, failing to account for the vast differences in spatio-temporal complexity among data samples Fan et al. (2021c). Consequently, for dynamically simple inputs, these models impose unnecessary computational overhead and potentially introduce noise, while for dynamically complex scenes, they may lack the adaptive capacity to capture critical motion patterns. Specifically, while Mamba-based SSMs are highly promising for long-term dependency modeling, their current implementations often rely on relatively fixed processing pipelines, which may limit adaptation to the varying spatio-temporal

patterns of sparse point-cloud videos under large-scale or high-frame-rate settings. This is a general observation about the modeling paradigm rather than a limitation of any specific method. To address this fundamental limitation, we argue that future models must possess enhanced content-aware capabilities. We therefore propose an innovative lightweight module, the Dynamic Spatio-Temporal Adapter (DST-Adapter). Preceded by a specialized embedding structure that decouples spatial and temporal features, the DST-Adapter first utilizes a context encoder to perceive the input's spatio-temporal characteristics. Based on this context, it generates dynamic gating weights to decide upon and execute a weighted fusion of three parallel, functionally decoupled feature streams: original, spatially enhanced, and temporally enhanced features. This allows the model to adaptively adjust its computational strategy, flexibly and efficiently fusing spatio-temporal information based on the intrinsic complexity of the input data.

This design allows DST-Adapter to not only provide customized processing strategies across diverse scenarios but also intelligently allocate computational resources, prioritizing the most critical spatio-temporal features. Experimental results on several public datasets demonstrate that our dynamic adaptation mechanism significantly enhances model performance in 4D point cloud video understanding tasks, offering a novel and more adaptive approach to spatio-temporal modeling.

Overall, our main contributions are as follows:

- The decoupling Spatio-Temporal Embedding Structure possesses stronger long-term dynamic perception capabilities. By generating decoupled temporal and spatial features within the embedding layer, it avoids the complexity of full-process modeling. This is the first module that combines long-term dynamic dependency modeling with efficient feature representation.

- We propose a innovative lightweight module, the Dynamic Spatiotemporal Adapter (DST-Adapter). DST-Adapter dynamically perceives the spatiotemporal features of the input data through a context encoder and adaptively generates dynamic gating weights based on different scenes, using FiLM modulation to control the contribution of three parallel feature streams.

- We conducted experiments on several publicly available 4D point cloud video recogintion benchmarks to validate the effectiveness of DST-Adapter. The experimental results demonstrate that the dynamic adaptation approach significantly improves the model's performance in 4D point cloud video understanding tasks, providing a new and more adaptive modeling paradigm for the field.

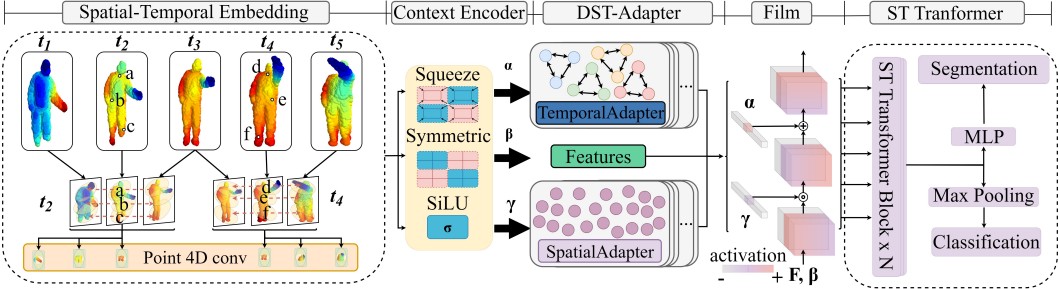

Figure 1: Pipeline. The backbone comprises four components: Spatio-Temporal Embedding, Context Encoder, DST-Adapter with FiLM, and Spatio-Temporal Transformer. Specifically, the "Squeeze" operation aggregates global spatial information to capture the holistic context, while the "Symmetric" operation enhances structural robustness by leveraging feature symmetry; collectively, they generate the parameters $(\alpha, \beta, \gamma)$ to guide subsequent feature modulation.

## 2  RELATED WORKS

**CNN-based 4D Point Cloud Understanding Backbones.** Understanding 4D point cloud videos is a challenging frontier in the field of 3D vision (Choe et al., 2022; Zheng et al., 2020; 2023; Liu et al., 2023b; Wiesmann et al., 2022; Qian et al., 2022). Early approaches for processing dynamic point

clouds primarily relied on CNN architectures (Luo et al., 2018; Guo et al., 2016; Wu et al., 2015; Qi et al., 2017a;b). These methods typically encode spatio-temporal information implicitly through carefully designed convolution operations, thereby avoiding the difficulties of explicit point tracking caused by inconsistencies across frames (Fan & Yang, 2019; Su et al., 2015; Maturana & Scherer, 2015; Qi et al., 2016; Saha et al., 2022). For example, MeteorNet appends a temporal dimension to 3D points and extends PointNet++ to track point motion (Zhong et al., 2022; Wang et al., 2019). PSTNet introduces an innovative Point Spatio-Temporal Convolution, which disentangles spatial and temporal processing: it first applies spatial convolution and then temporal convolution to the extracted spatial region features, thereby capturing dynamics while mitigating the impact of spatial irregularity Fan et al. (2021b). However, while CNN-based methods are effective at local feature extraction, their fixed receptive fields and static computational graphs limit their adaptability to complex dynamic scenarios, making it difficult to flexibly adjust computational strategies according to the spatio-temporal complexity of the input.

**Transformer-based 4D Point Cloud Understanding Backbones.** To overcome the local receptive field limitations of CNNs, researchers introduce Transformers into 4D point cloud understanding Zhang et al. (2023); Liu et al. (2023a); Shen et al. (2023a;b). P4Transformer is a pioneering work in this direction, combining convolution with self-attention for spatio-temporal modeling Fan et al. (2021a), but it incurs prohibitively high computational costs. PPTr alleviates this issue by leveraging primitive planes as compact mid-level representations Wen et al. (2022). PST-Transformer further improves the design by explicitly integrating decoupled spatio-temporal encoding into its attention mechanism, addressing the lack of explicit structure modeling in P4Transformer Fan et al. (2022). Despite these advancements, Transformer-based methods still rely on static attention mechanisms and uniform computational pipelines. They treat all inputs equally, regardless of their intrinsic temporal dynamics, which can result in wasted computation for simple sequences and insufficient modeling for complex ones.

**Recent Advancements in Architecture.** In recent studies, improving computational efficiency has become a major trend in 4D point cloud understanding. One line of work explores State Space Models (SSMs) as efficient alternatives to computationally expensive Transformers. A representative example is Mamba4D, which employs intra-frame spatial Mamba and inter-frame temporal Mamba to capture short- and long-range dependencies, achieving linear complexity for long-sequence modeling (Liu et al., 2025; Gu & Dao, 2023; Liu et al., 2024; Yu & Wang, 2024; Li et al., 2024; Liang et al., 2024; Oshima et al., 2024). Another line of work focuses on operator design. For instance, PvNeXt introduces a Motion Imitator to generate "virtual motion frames" and employs a one-shot query mechanism to replace traditional dense frame-by-frame neighborhood querying, thereby significantly reducing computational overhead (Wang et al., 2025). These studies collectively reflect the trend of improving efficiency through operator redesign. However, they still rely on fixed computational graphs and uniform processing strategies, overlooking the diverse spatio-temporal complexity of different inputs. This one-size-fits-all paradigm limits adaptability, often leading to redundant computation or insufficient modeling.

In summary, CNN, Transformer, and SSM-based backbones have advanced 4D point cloud understanding, but they all rely on static and uniform computation, overlooking variations in spatio-temporal complexity. To address this limitation, we propose a Spatio-Temporal Decoupling Embedding Structure and a dynamic, content-aware mechanism that adaptively adjusts computation, offering a more flexible and effective paradigm for 4D point cloud video understanding.

## 3 METHODOLOGY

In this section, we introduce our framework for content-aware 4D point cloud processing, centered around a novel Pre-Embedding Dynamic Spatio-Temporal Decoupling approach (illustrated in Figure 1). We begin in Section 3.1 by analyzing the task and formulating the problem. Subsequently, we dissect the core components of our proposed DST-Adapter: Section 3.2 details the Spatio-Temporal Tube Embedding for compact feature representation. Section 3.3 elaborates on the Decoupled Spatio-Temporal Adapters, which process features through distinct identity, spatial, and temporal branches. Section 3.4 describes the Dynamic Gating and Fusion Mechanism that enables adaptive feature integration. In Section 3.5, we present the FiLM-based Feature Modulation for global context refinement. Finally, Section 3.6 provides a formal mathematical summary of the entire method.

## 3.1 TASK ANALYSIS AND PROBLEM MODELING

We first formulate the representation of a 4D point cloud video sample.

$$x = \{X_t \in \mathbb{R}^{(3+C) \times N}\}_{t=1}^{T}, \tag{1}$$

where $N$ is the number of points per frame, $T$ is the number of frames, 3 is the spatial coordinate dimension, and $C$ is the point-wise feature dimension. The goal is to learn a mapping from such spatio-temporal sequences to task-specific outputs (e.g., action labels or per-point semantic labels) under strong inter-frame inconsistency, spatial disorder, long-range temporal dependencies, and heterogeneous spatio-temporal complexity across videos.

We introduce a novel content-aware 4D point cloud processing approach, termed DSTA4D that decouples spatial and temporal representations within the embedding layer, and a Dynamic Spatio-Temporal Adapter (DST-Adapter) that adaptively fuses three parallel branches (identity, spatial-enhanced, temporal-enhanced) with content-aware gating and FiLM modulation. This aims to both avoid the complexity of full-process long-term modeling and allocate computation adaptively to the most critical dimensions conditioned on the input video.

## 3.2 CONTEXT ENCODER

To achieve content awareness, the model must first establish a global understanding of the input's overall characteristics. The role of the Context Encoder is to compress the input feature embeddings $\boldsymbol{E}_{\text{base}} \in \mathbb{R}^{B \times T \times N \times C}$, where $B$ is the batch size, $T$ is the number of temporal frames, $N$ is the number of points per frame, and $C$ is the feature dimension, into a compact global context vector $\boldsymbol{ctx} \in \mathbb{R}^{B \times D_{\text{ctx}}}$, with $D_{\text{ctx}}$ denoting the dimension of the context vector. This process is designed to capture a comprehensive and rich statistical descriptor through two key steps:

**Multi-dimensional, Multi-modal Pooling.** We creatively employ both mean and max pooling modes across both temporal and spatial dimensions. Mean pooling captures the "average" or "general" state of the sequence, such as the mean posture of an action. In contrast, max pooling captures the "peak" or "most salient" features, like the extremities of a motion trajectory or moments of maximum velocity. This combination provides a far more discriminative descriptor for distinguishing between subtle and vigorous actions than using global average pooling alone.

**Global Information Fusion and Refinement.** After concatenating the four statistical descriptors derived from different perspectives, we use a small MLP for non-linear projection. This step is crucial as it learns to intelligently combine and refine these raw statistics into a more abstract and useful final context representation, $\boldsymbol{ctx}$, for downstream tasks. The $\boldsymbol{ctx}$ vector serves as a global descriptor ("fingerprint") of the input sequence, capturing its overall motion intensity and spatial complexity.

$$\boldsymbol{S} = \text{Concat}[\boldsymbol{a}, \boldsymbol{b}, \boldsymbol{c}, \boldsymbol{d}] \in \mathbb{R}^{B \times 4C}, \qquad \boldsymbol{ctx} = \text{MLP}_{\text{proj}}(\boldsymbol{S}) \in \mathbb{R}^{B \times D_{\text{ctx}}}, \tag{2}$$

we further clarify that the dynamic gating weights $(\alpha, \beta, \gamma)$ used in the DST-Adapter are generated from this context vector through an additional lightweight prediction head.

$$[\alpha, \beta, \gamma] = \sigma\big(\text{MLP}_{\text{gate}}(\boldsymbol{ctx})\big) \in \mathbb{R}^{B \times 3}, \tag{3}$$

where $B$ is the batch size, $C$ is the feature dimension, and $D_{\text{ctx}}$ denotes the dimension of the context vector. Here, $\boldsymbol{a}$ and $\boldsymbol{b}$ represent the temporal mean and max pooled features, while $\boldsymbol{c}$ and $\boldsymbol{d}$ represent the spatial mean and max pooled features, respectively. The gating weights $(\alpha, \beta, \gamma)$ are therefore context-aware coefficients derived from $\boldsymbol{ctx}$ to modulate the three branches of the DST-Adapter dynamically.

## 3.3 DECOUPLED SPATIO-TEMPORAL ADAPTERS

While the global context is being derived, the main feature stream $\boldsymbol{E}_{\text{base}}$ is processed in parallel through three functionally decoupled branches:

**Identity Path.** This path directly preserves the original features $\boldsymbol{E}_{\text{base}}$. It serves as a conservative option, ensuring that the model can fall back on the base features when spatio-temporal information is not prominent, thus preventing the introduction of noise from unnecessary transformations.

**Spatial Adapter.** This module focuses on enhancing intra-frame spatial feature interactions, reflecting the principle of orthogonal functional decomposition. It comprises two parallel sub-paths: Token-Mixing, responsible for learning relationships among different points to capture local geometry, and Channel-Mixing, which learns relationships within the feature channels of a single point via a parameter-efficient Low-Rank Adapter (LoRAAdapter). This parallel processing of "external relationships" and "internal representation" facilitates a more comprehensive learning of spatial features.

**Temporal Adapter.** This module is dedicated to capturing inter-frame temporal dynamics. It employs a lightweight Depthwise 1D Convolution that slides along the temporal axis $T$. This design is both efficient and effective for capturing local motion patterns and short-term trajectory information.

## 3.4 DYNAMIC GATING AND FUSION MECHANISM

This is the decision-making core of the DST-Adapter. The context vector $\boldsymbol{ctx}$ is fed into a small gating network to dynamically generate normalized weights $(\alpha, \beta, \gamma)$ for the three parallel feature branches.

$$[\alpha, \beta, \gamma] = \text{Softmax}(\text{MLP}_{\text{gate}}(\boldsymbol{ctx})) \tag{4}$$

The fusion process is a dynamic weighted sum:

$$\boldsymbol{E}_{\text{fused}} = \alpha \cdot \boldsymbol{E}_{\text{base}} + \beta \cdot \boldsymbol{E}_s + \gamma \cdot \boldsymbol{E}_t \tag{5}$$

where, $\boldsymbol{E}_s$ denotes the spatially enhanced features from the Spatial Adapter, and $\boldsymbol{E}_t$ denotes the temporally enhanced features from the Temporal Adapter. This mechanism enables the model to intelligently allocate computational resources based on the global context, transforming the fusion from a static operation into a flexible, content-adaptive process.

## 3.5 FEATURE MODULATION WITH FiLM

Following the dynamic fusion, we introduce a Feature-wise Linear Modulation (FiLM) layer as a final refinement step. FiLM can be viewed as a global context-based post-processor. It utilizes the $\boldsymbol{ctx}$ vector to generate a unique scaling factor $\boldsymbol{\sigma}$ and a shifting offset $\boldsymbol{\delta}$ for each feature channel.

$$\boldsymbol{\sigma} = \text{Linear}_{\text{scale}}(\boldsymbol{ctx}) \in \mathbb{R}^{B \times C} \quad \boldsymbol{\delta} = \text{Linear}_{\text{shift}}(\boldsymbol{ctx}) \in \mathbb{R}^{B \times C} \tag{6}$$

The fused features $\boldsymbol{E}_{\text{fused}}$ then undergo an affine transformation to produce the final output $\boldsymbol{E}_{\text{adapted}}$:

$$\boldsymbol{E}_{\text{adapted}} = \boldsymbol{E}_{\text{fused}} \odot (1 + \tanh(\boldsymbol{\sigma})) + \boldsymbol{\delta} \tag{7}$$

If dynamic gating decides *which* expert's opinion to weigh more, the FiLM layer performs an overall *recalibration* of the fused opinion based on the global situation, further enhancing the discriminative power of the features.

## 3.6 METHOD FORMALIZATION

Let the base embedding be $\mathbf{E}_{\text{base}} \in \mathbb{R}^{T \times N \times C}$. The pre-ST embedding produces spatial and temporal features

$$\mathbf{z}_s = \phi_s(\mathbf{E}_{\text{base}}), \qquad \mathbf{z}_t = \phi_t(\mathbf{E}_{\text{base}}), \tag{8}$$

where $\phi_s$ uses local spatial neighborhoods and $\phi_t$ uses inter-frame dynamics.

DST-Adapter constructs three parallel streams:

$$\mathbf{f}_0(x) = \mathbf{E}_{\text{base}}, \quad \mathbf{f}_s(x) = S(\mathbf{E}_{\text{base}}), \quad \mathbf{f}_t(x) = T(\mathbf{E}_{\text{base}}). \tag{9}$$

A context encoder yields $\text{ctx}(x)$ and the gating network $g$ outputs

$$[\alpha(x), \beta(x), \gamma(x)] = \mathrm{softmax}\big(g(\mathrm{ctx}(x))\big) \in \Delta^2. \qquad (10)$$

The fused feature and FiLM-conditioned output are

$$\mathbf{E}_{\mathrm{fused}}(x) = \alpha\mathbf{f}_0 + \beta\mathbf{f}_s + \gamma\mathbf{f}_t, \qquad \mathbf{E}_{\mathrm{adapted}}(x) = \mathrm{FiLM}\big(\mathbf{E}_{\mathrm{fused}}; \sigma(x), \delta(x)\big), \qquad (11)$$

with per-channel FiLM parameters $(\sigma, \delta)$ predicted from $\mathrm{ctx}(x)$. A task head $h$ produces $\hat{y} = h(\mathbf{E}_{\mathrm{adapted}})$, optimized under loss $\ell(\hat{y}, y)$.

# 4 EXPERIMENTS

## 4.1 EXPERIMENTAL SETUP

### 4.1.1 DATASETS

We conduct experiments on three widely-used benchmark datasets: MSR-Action3D, NTU RGBD for 3D action recognition and Synthia 4D for 4D semantic segmentation (Shahroudy et al., 2016; Li et al., 2010; Choy et al., 2019).

**MSR-Action3D.** For our empirical evaluation, we employ a widely-adopted dataset composed of 567 depth videos recorded with a first-generation Kinect sensor. This collection provides a diverse set of 20 human action classes, with a cumulative frame count of roughly 23,000 and a mean video length of 40 frames. To ensure a fair and direct comparison with prior works, we follow the conventional data partition, utilizing 270 sequences for model training and 297 for performance testing.

**NTU RGBD.** NTU RGBD dataset is a large-scale benchmark for 3D human action recognition. It contains a total of 56,880 videos distributed across 60 fine-grained action categories. The length of each video varies, ranging from approximately 30 to 300 frames. For the standard cross-subject evaluation protocol, the dataset is split into 40,320 videos for training and 16,560 videos for testing.

**Synthia 4D.** To assess the generalization capability of our model on dense prediction tasks, we further conduct 4D semantic segmentation experiments on the Synthia 4D dataset. This dataset, derived from Synthia (Ros et al., 2016), consists of six dynamic driving scenarios. We adhere to the established experimental protocol from prior works (Fan et al., 2021a), utilizing a standard frame-wise split of 19,888 for training, 815 for validation, and 1,886 for testing.

### 4.1.2 TRAINING DETAILS

**MSR-Action3D.** We follow P4Transformer to partition the training and testing sets. For each video, we densely sample 24 frames and sample 2048 points in each frame. We train our model on a single NVIDIA A100 GPU for 50 epochs. The SGD optimizer is employed, where the initial learning rate is set as 0.01, and decays with a rate of 0.1 at the 20-th epoch and the 30-th epoch respectively.

Table 1: Action recognition accuracy comparison (%) on MSR-Action3D.

| Methods | Input | Acc (%) |
|---|---|---|
| PSTNet(Fan et al., 2021b) | point | 91.20 |
| P4Transformer(Fan et al., 2021a) | point | 90.94 |
| PSTNet++(Fan et al., 2021c) | point | 92.68 |
| PST-Transformer(Fan et al., 2022) | point | 93.73 |
| PPTr+C2P(Zhang et al., 2023) | point | 94.76 |
| PointCPSC(Shen et al., 2023b) | point | 92.68 |
| MaST-Pre(Shen et al., 2023a) | point | 94.08 |
| SequentialPointNet(Li et al., 2022) | point | 92.64 |
| MAMBA4D(Liu et al., 2025) | point | 93.38 |
| 3DInAction(Ben-Shabat et al., 2024) | skeleton | 92.23 |
| PvNeXt(Wang et al., 2025) | point | 94.77 |
| DSTA4D(Ours) | point | **96.17** |

**NTU RGBD.** Consistent with prior research, the model processes 24 frames at each training step, each containing 2048 points, with a temporal step of 2. This dataset's training extends over 15 epochs with a batch size of 24, utilizing a single NVIDIA A100 GPU. The SGD optimizer is employed, setting

the initial learning rate at 0.01 with a cosine decay. This configuration ensures stable convergence and fair comparison with previous works.

**Synthia 4D.** For model optimization on the Synthia 4D dataset, we employ the SGD optimizer with an initial learning rate of 0.01 and a momentum of 0.9. The learning rate is managed by a multi-step schedule, which is preceded by a 10-epoch linear warmup phase. Following the warmup, the learning rate is decayed by a factor of 0.1 at the 50th, 80th, and 100th epochs. Our model is trained for a total of 150 epochs, processing input as 3-frame clips with 16,384 points per frame, using a batch size of 8. All experiments were conducted on a single NVIDIA A100 GPU.

## 4.2 MSR-ACTION3D

**Quantitative results.** Table 1 presents a comprehensive performance comparison of our proposed method against several state-of-the-art approaches on the action recognition benchmark. As demonstrated in the results, our method achieves a new state-of-the-art accuracy of 96.17%. This marks a significant improvement of 1.40 percentage points over the next-best performing method, PvNeXt (94.77%), and a 1.41 point lead over PPTr+C2P (94.76%). Furthermore, when compared to other recent and highly relevant models such as PST-Transformer + MaST-Pre (94.08%) and MAMBA4D (93.38%), our approach consistently shows a clear superiority. These results strongly validate the effectiveness and advanced capabilities of our proposed architecture in capturing complex spatio-temporal dynamics from point cloud sequences.

Table 2: Action recognition accuracy (%) on NTU RGBD.

| Method | Input | Acc (%) |
|---|---|---|
| SkeleMotion(Caetano et al., 2019) | skeleton | 69.6 |
| GCA-LSTM(Liu et al., 2017b) | skeleton | 74.4 |
| AttentionLSTM(Liu et al., 2017a) | skeleton | 77.1 |
| AGC-LSTM(Si et al., 2019) | skeleton | 89.2 |
| AS-GCN(Li et al., 2019) | skeleton | 86.8 |
| VA-fusion(Zhang et al., 2019) | skeleton | 89.4 |
| 2s-AGCN(Shi et al., 2019b) | skeleton | 88.5 |
| DGNN(Shi et al., 2019a) | skeleton | 89.9 |
| HON4D(Oreifej & Liu, 2013) | depth | 30.6 |
| SNV(Yang & Tian, 2014) | depth | 31.8 |
| HOG$^2$(Ohn-Bar & Trivedi, 2013) | depth | 32.2 |
| Li et al.(Li et al., 2018) | depth | 68.1 |
| Wang et al.(Wang et al., 2018) | depth | 87.1 |
| MVDI(Xiao et al., 2019) | depth | 84.6 |
| PointNet++(Qi et al., 2017b) | point | 80.1 |
| 3DV(Wang et al., 2020) | voxel | 84.5 |
| 3DV-PointNet++(Wang et al., 2020) | voxel+point | 88.8 |
| PSTNet(Fan et al., 2021b) | point | 90.5 |
| P4Transformer(Fan et al., 2021a) | point | 90.2 |
| PST-Transformer(Fan et al., 2022) | point | 91.0 |
| SequentialPointNet(Li et al., 2022) | point | 90.3 |
| MaST-Pre(Shen et al., 2023a) | point | 90.8 |
| PvNeXt(Wang et al., 2025) | point | 89.2 |
| DSTA4D(Ours) | point | **91.3** |

## 4.3 NTU RGBD

**Quantitative results.** As reported in Table 2, our method achieves state-of-the-art performance on the NTU RGB+D dataset. Under the Cross-Subject protocol, our model obtains an accuracy of 91.3%, surpassing all listed competitors. This result represents a significant improvement over strong point-based and voxel-based baselines. Notably, our model outperforms PST-Transformer (91.0%) and achieves a 2.5 percentage point gain over 3DV-PointNet++ (88.8%). Unlike approaches that depend on motion extraction from voxelized inputs, our framework directly captures spatio-temporal dynamics from raw point cloud sequences, leading to superior accuracy. These results highlight the effectiveness and robustness of our approach for large-scale 3D action recognition.

## 4.4 4D SEMANTIC SEGMENTATION

**Quantitative results.** As delineated in Table 3, we present a comprehensive comparison of our model against various state-of-the-art methods for 4D semantic segmentation on the Synthia 4D dataset. Our approach is evaluated in both single-frame (frame=1) and multi-frame (frame=3) settings to demonstrate its effectiveness and temporal modeling capabilities.In the more challenging multi-

Table 3: 4D semantic segmentation results (mIoU %) on the Synthia 4D dataset.

| Method | Input | Frame | Track | Bldn | Road | Sdwlk | Fence | Vegittn | Pole | - |
|---|---|---|---|---|---|---|---|---|---|---|
| 3D MinkNet14 | voxel | 1 | - | 89.39 | 97.68 | 69.43 | 86.52 | 98.11 | 97.26 | - |
| 4D MinkNet14 | voxel | 3 | - | 90.13 | 98.26 | 73.47 | 87.19 | 99.10 | 97.50 | - |
| PointNet++ | point | 1 | - | 96.88 | 97.72 | 86.20 | 92.75 | 97.12 | 97.09 | - |
| MeteorNet-m | point | 2 | ✓ | **98.22** | 97.79 | 90.98 | 93.18 | 98.31 | 97.45 | - |
| MeteorNet-m | point | 2 | × | 97.65 | 97.83 | 90.03 | 94.06 | 97.41 | 97.79 | - |
| MeteorNet-l | point | 3 | × | 98.10 | 97.72 | 88.65 | 94.00 | 97.98 | 97.65 | - |
| P4Transformer | point | 1 | - | 96.76 | 98.23 | 92.11 | 95.23 | 98.62 | 97.77 | - |
| P4Transformer | point | 3 | × | 96.73 | 98.35 | 94.03 | 95.23 | 98.28 | 98.01 | - |
| PST-Transformer | point | 1 | - | 94.46 | 98.13 | 89.37 | 95.84 | 99.06 | 98.10 | - |
| PST-Transformer | point | 3 | × | 96.10 | 98.44 | 94.94 | **96.58** | 98.98 | 98.10 | - |
| MAMBA4D | point | 3 | × | 96.16 | **98.58** | 92.80 | 94.95 | 97.08 | 98.24 | - |
| DSTA4D(Ours) | point | 1 | - | 96.99 | 98.57 | 94.51 | 95.61 | 98.78 | **98.30** | - |
| DSTA4D(Ours) | point | 3 | × | 97.30 | 98.56 | **95.25** | 95.86 | **99.51** | 98.22 | - |

| Method | Input | Frame | Track | Car | T. Sign | Pedstrn | Bicycl | Lane | T. Light | mIoU |
|---|---|---|---|---|---|---|---|---|---|---|
| 3D MinkNet14 | voxel | 1 | - | 93.50 | 79.45 | 92.27 | 0.00 | 44.61 | 66.69 | 76.24 |
| 4D MinkNet14 | voxel | 3 | - | 94.01 | 79.04 | 92.62 | 0.00 | 50.01 | 68.14 | 77.46 |
| PointNet++ | point | 1 | - | 90.85 | 66.87 | 78.64 | 0.00 | 72.93 | 75.17 | 79.35 |
| MeteorNet-m | point | 2 | ✓ | 94.30 | 76.35 | 81.05 | 0.00 | 74.09 | 75.92 | 81.47 |
| MeteorNet-m | point | 2 | × | 94.15 | 82.01 | 79.14 | 0.00 | 72.59 | 77.92 | 81.72 |
| MeteorNet-l | point | 3 | × | 93.83 | 84.07 | 80.90 | 0.00 | 71.14 | 77.60 | 81.80 |
| P4Transformer | point | 1 | - | 95.46 | 80.75 | 85.48 | 0.00 | 74.28 | 74.22 | 82.41 |
| P4Transformer | point | 3 | × | 95.60 | 81.54 | 85.18 | 0.00 | 75.95 | 79.07 | 83.16 |
| PST-Transformer | point | 1 | - | **96.80** | 80.41 | 87.58 | 0.00 | 75.25 | 80.84 | 82.92 |
| PST-Transformer | point | 3 | × | 96.06 | 82.67 | 87.86 | 0.00 | 76.01 | 81.67 | 83.95 |
| MAMBA4D | point | 3 | × | 95.75 | 82.03 | 84.57 | 0.00 | **79.35** | 80.74 | 83.35 |
| DSTA4D(Ours) | point | 1 | - | 95.80 | 83.96 | 87.03 | 0.00 | 77.55 | 80.81 | 83.99 |
| DSTA4D(Ours) | point | 3 | × | 96.37 | **85.75** | **90.46** | 0.00 | 77.19 | **82.05** | **84.71** |

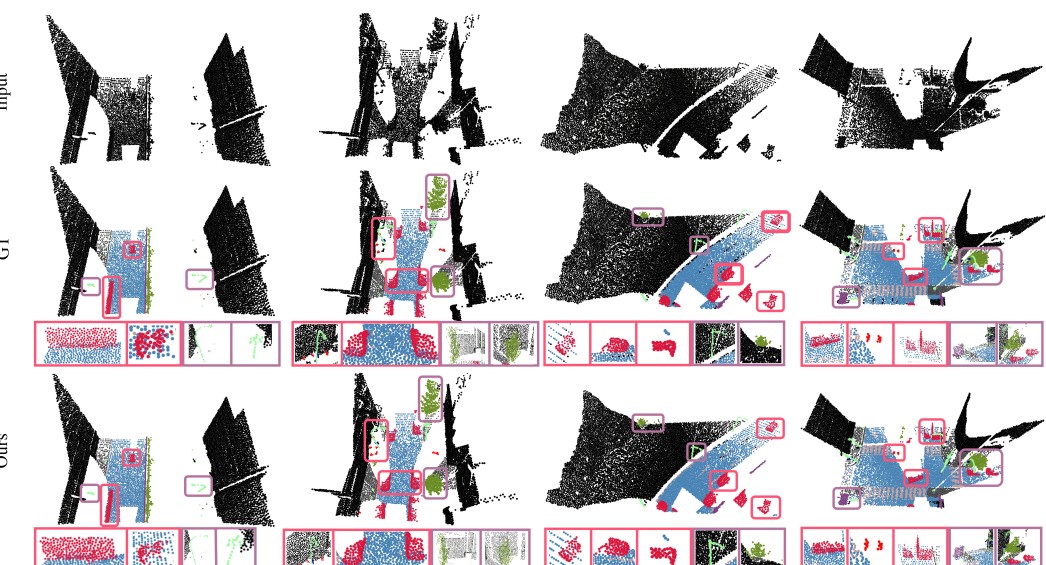

Figure 2: 4D Qualitative Results. The rows from top to bottom correspond to the input, GT, and our predictions. Detailed comparative results are highlighted in the enlarged regions of the figure.

frame setting, our model (Ours (frame=3)) achieves a new state-of-the-art performance, attaining a final mIoU of 84.71%. This result surpasses the next-best method, PST-Transformer (83.95%), by a margin of 0.76 percentage points. Notably, our model demonstrates superior segmentation performance across several difficult categories, securing the top scores for Sidewalk (95.25%),

Table 4: Ablation studies on MSR-Action3D.

| Model | Acc(%) |
|---|---|
| Full Model | 96.17 |
| w/o Film | 94.08 |
| w/o Decoupled Spatio-Temporal Adapters | 93.37 |

Table 5: Comparison results with advanced architecture processing more frames.

| Method | Backbone | Frame | Acc(%) |
|---|---|---|---|
| PvNeXt | CNN | 24 | 94.77 |
| MAMBA4D | Mamba | 32 | 93.38 |
| Ours | Transformer | 24 | 96.17 |

Vegetation (99.51%), Traffic Sign (85.75%), Pedestrian (90.46%), and Traffic Light (82.05%).In the single-frame setting, our model (Ours (frame=1)) also establishes its superiority by achieving an mIoU of 83.99%, outperforming all other single-frame competitors, including the strong PST-Transformer baseline (82.92%). Furthermore, the performance gain from our single-frame to multi-frame variant highlights our model's robust ability to effectively leverage temporal information for more accurate segmentation.These comprehensive results across both settings and multiple semantic classes firmly establish the superiority and robustness of our proposed architecture.

**Qualitative results.** To visually assess our model, Fig. 2 compares the raw input, ground-truth (GT), and our predictions. The results show high consistency with GT across diverse scenes. Even in challenging cases with complex boundaries and fine details, our model produces accurate segmentation maps, demonstrating its robustness and strong generalization ability.

## 4.5 ABLATION STUDIES

We conduct ablation experiments on the MSR-Action3D dataset by systematically removing each core module and comparing with the full model. As shown in Table 4, the results confirm that all components are essential to the model's overall performance.

**Comparative Study on Model Effectiveness and Efficiency.** To further benchmark our model against advanced architectures designed for extended temporal data, we introduce two representative methods: the Mamba-based MAMBA4D and the CNN-based PvNeXt. As detailed in Table 5, MAMBA4D leverages its long-sequence-optimized structure to process 32 frames, achieving an accuracy of 93.38%. Concurrently, the efficient PvNeXt model attains a competitive result of 94.77% using 24 frames. Notably, our proposed Transformer-based model achieves a superior accuracy of 96.17% with the same 24-frame input, significantly outperforming both counterparts. This result strongly demonstrates our model's exceptional capability in effectively capturing and utilizing spatio-temporal dynamics, striking a superior balance between performance and efficiency.

**Effect of temporal kernel size and spatial radius.** We conducted an ablation study on the model's temporal kernel size and spatial radius in Fig 3. The results indicate that the model achieves optimal recognition accuracy when the temporal kernel size is set to 3 and the spatial radius is set to 0.3. Specifically, performance degrades as the temporal kernel size increases, while the spatial radius requires a balanced value, as both excessively large and small radii are detrimental to model performance.

**Addition qualitative results.** Furthermore, we present two segmentation results from the Synthia 4D dataset in Fig. 4. The experiments demonstrate that our method achieves accurate segmentation for most objects while producing fewer incorrect predictions compared to other approaches.

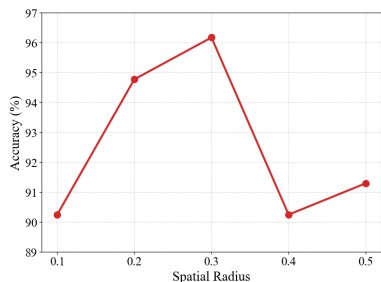

(a) Temporal kernel sizes

(b) Spatial radius settings

Figure 3: Analysis of accuracy with varying temporal and spatial parameters.

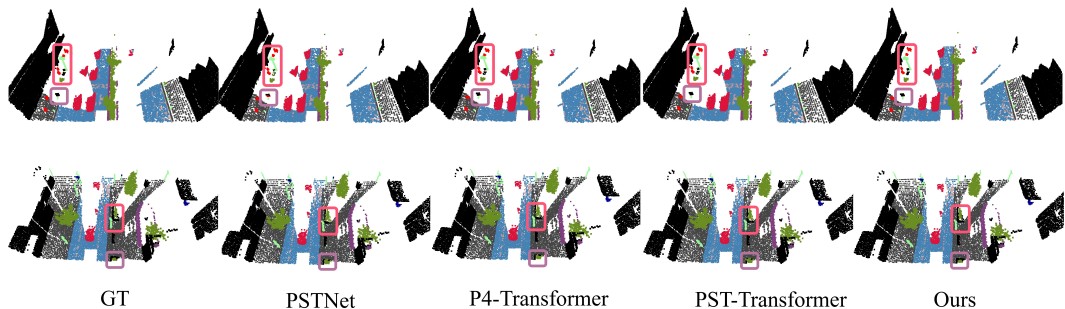

Figure 4: Additional 4D semantic segmentation visualization results on the Synthia 4D dataset. Highlighted regions indicate comparative outcomes, where our DSTA4D method shows the closest similarity to the ground truth (GT) and produces fewer incorrect predictions than other approaches.

## 5 CONCLUSION

This study proposes an innovative pre-embedding-based dynamic spatio-temporal decoupling method for content-aware 4D point cloud processing, which can efficiently handle the complex spatio-temporal features in 4D point cloud videos. By introducing the pre-embedding structure and Dynamic Spatio-Temporal Adapter (DST-Adapter), we have successfully optimized the processing flow of spatio-temporal features, significantly enhancing the model's ability to model long-term dependencies and adapt to different environments. Experimental results on mainstream benchmark tests show that the proposed method significantly improves performance, providing a more efficient and intelligent adaptive modeling paradigm for the field of point cloud video understanding.

## 6 REPRODUCIBILITY STATEMENT

We provide full access to the experimental data, source code, and pretrained models through anonymous links in the supplemental material. We describe the experimental environment in detail, including operating system version, Python version, dependencies, and the exact command-line scripts used to run the experiments. We also include detailed instructions for accessing and pre-processing the raw data. These resources ensure that other researchers can accurately reproduce all reported results. Anonymous open-source code repository: `https://anonymous.4open.science/r/DSTA4d-8D5D`

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

## A  APPENDIX

### A.1  THEORETICAL RESULTS

We use standard assumptions: $\ell(\cdot, y)$ is convex and $L$-Lipschitz; the gating and branch families have sufficient capacity; FiLM scalings are bounded to ensure a global Lipschitz constant.

**Conditional mixture is not worse than the best static convex fusion.** Let $w^\star \in \Delta^2$ be the optimal static convex weight minimizing

$$\mathbb{E}\big[\ell\big(w_0 \mathbf{f}_0(x) + w_s \mathbf{f}_s(x) + w_t \mathbf{f}_t(x),\, y\big)\big]. \tag{12}$$

There exists a gating parameterization such that the DST-Adapter output

$$\mathbf{f}_{\mathrm{DST}}(x) = w_0(x)\mathbf{f}_0(x) + w_s(x)\mathbf{f}_s(x) + w_t(x)\mathbf{f}_t(x), \quad w(x) \in \Delta^2, \tag{13}$$

satisfies

$$\mathbb{E}\big[\ell(\mathbf{f}_{\mathrm{DST}}(x), y)\big] \leq \mathbb{E}\big[\ell\big(w_0^\star \mathbf{f}_0 + w_s^\star \mathbf{f}_s + w_t^\star \mathbf{f}_t,\, y\big)\big]. \tag{14}$$

Choose the gating to be constant: $w(x) \equiv w^\star$. Then $\mathbf{f}_{\mathrm{DST}}$ equals the best static convex fusion and achieves the same risk. If the data distribution is heterogeneous, allowing $w(x)$ to vary with $x$ can only reduce the expected loss further by convexity and Jensen's inequality.

Oracle inequality for per-sample best expert. Let $k^\star(x) = \arg\min_{k \in \{0, s, t\}} \ell(\mathbf{f}_k(x), y)$. Suppose the gating top-1 selection matches $k^\star(x)$ with probability at least $p > 1/3$. If $\ell$ is $L$-Lipschitz and $\mathbb{E}\|\mathbf{f}_k(x) - \mathbf{f}_{k^\star(x)}(x)\| \leq D$, then

$$\mathbb{E}[\ell(\mathbf{f}_{\mathrm{DST}}(x), y)] \leq \mathbb{E}\big[\ell(\mathbf{f}_{k^\star(x)}(x), y)\big] + (1 - p)\, L\, D. \tag{15}$$

Decompose the expectation over the events $\{\arg\max w(x) = k^\star(x)\}$ and its complement. When the best expert is selected, convexity ensures $\ell(\mathbf{f}_{\mathrm{DST}}) \leq \ell(\mathbf{f}_{k^\star})$. Otherwise, Lipschitz continuity yields $\ell(\mathbf{f}_{\mathrm{DST}}) - \ell(\mathbf{f}_{k^\star}) \leq L\|\mathbf{f}_{\mathrm{DST}} - \mathbf{f}_{k^\star}\| \leq LD$ in expectation. Weighting by probabilities gives the result.

**Variance reduction via embedding-layer spatio-temporal decoupling** Assume the target representation decomposes as $\mathbf{z} = \phi_s(x) + \phi_t(x)$ and the estimators $\hat{\phi}_s, \hat{\phi}_t$ are obtained in the embedding layer under a structural decoupling constraint that enforces near-orthogonality of errors: $\mathbb{E}\langle \hat{\phi}_s - \phi_s, \hat{\phi}_t - \phi_t \rangle \approx 0$. Then

$$\mathbb{E}\|\hat{\mathbf{z}} - \mathbf{z}\|^2 = \mathbb{E}\|\hat{\phi}_s - \phi_s\|^2 + \mathbb{E}\|\hat{\phi}_t - \phi_t\|^2 \pm \varepsilon, \tag{16}$$

with a small cross-term $\varepsilon \approx 0$. Furthermore, with DST gating $g_s(x), g_t(x)$,

$$\mathbb{E}\|g_s(\hat{\phi}_s - \phi_s) + g_t(\hat{\phi}_t - \phi_t)\|^2 \leq \mathbb{E}[g_s^2] \cdot \mathrm{Var}_s + \mathbb{E}[g_t^2] \cdot \mathrm{Var}_t + \tilde{\varepsilon}, \tag{17}$$

so that the overall variance is reduced when the data distribution is heterogeneous across "spatially-dominant" and "temporally-dominant" subpopulations.

The first identity follows from expanding the squared norm and using the near-orthogonality assumption, which can be enforced by architectural separation and/or orthogonalization penalties in the embedding layer. For the gated estimator, expand the squared norm and take expectations; the mixed term is suppressed by the same structural decoupling, while the factors $\mathbb{E}[g_s^2], \mathbb{E}[g_t^2]$ downscale the irrelevant-branch variance on each subpopulation, yielding a strictly smaller global variance when heterogeneity is present.

**Content-aware compute allocation is first-order optimal under a budget.** Let per-branch costs be $c_0, c_s, c_t$ and define the per-sample cost $c(x) = \alpha c_0 + \beta c_s + \gamma c_t$. Consider

$$\min_{\alpha(x),\beta(x),\gamma(x)\in\Delta^2} \; \mathbb{E}[\ell(\mathbf{f}_{\mathrm{DST}}(x),y)] \quad \text{s.t.} \quad \mathbb{E}[c(x)] \leq B. \tag{18}$$

Then content-aware gating that increases $\beta(x)$ on spatially constrained subsets $\mathcal{X}_s$ where $\partial\ell/\partial\beta < 0$ and increases $\gamma(x)$ on temporally constrained subsets $\mathcal{X}_t$ where $\partial\ell/\partial\gamma < 0$ satisfies the KKT first-order necessary conditions, with the Lagrange multiplier balancing the global budget.

Form the Lagrangian $\mathcal{L} = \mathbb{E}[\ell] + \lambda\big(\mathbb{E}[c] - B\big)$ and differentiate w.r.t. $\alpha, \beta, \gamma$ under the simplex and budget constraints. On $\mathcal{X}_s$, $\partial\mathcal{L}/\partial\beta = \mathbb{E}[\partial\ell/\partial\beta] + \lambda c_s$ is minimized by increasing $\beta$ if $\partial\ell/\partial\beta < -\lambda c_s$; similarly for $\gamma$ on $\mathcal{X}_t$. The complementary slackness and feasibility conditions complete the KKT characterization.

**Expressivity and stability of FiLM-conditioned fusion.** Let $\mathrm{FiLM}(\mathbf{u}; \sigma, \delta) = (1 + \tanh\sigma)\odot\mathbf{u} + \delta$ with per-channel parameters predicted from $\mathrm{ctx}(x)$. Then: (i) The family $\{x \mapsto (1 + \tanh\sigma(x))\odot \mathbf{u}(x) + \delta(x)\}$ realizes a rich class of conditional affine transformations that strictly expands the expressivity of convex fusion without increasing depth; (ii) Since $(1 + \tanh\sigma)\in(0, 2)$, the global Lipschitz constant of the adapter remains bounded, improving numerical stability and preventing gradient explosion on long sequences.

(i) Universal approximation with conditional affine modulations is standard: composing a fixed backbone feature $\mathbf{u}(x)$ with input-dependent per-channel scaling and shifting strictly enlarges the representable function class beyond fixed convex combinations.

(ii) The bounded scaling ensures that the operator norm of the FiLM layer is uniformly bounded, which upper-bounds the Lipschitz constant of the adapter and stabilizes optimization.

**Generalization advantage via data-dependent effective complexity.** Let $\mathcal{H}$ be the hypothesis class of the static backbone and $\mathcal{H}_{\mathrm{eff}}(x)$ denote the data-dependent sub-class activated by the gating on input $x$. If for a large measure of "simple" inputs the gating suppresses high-complexity paths, the empirical local Rademacher complexity $\hat{\mathfrak{R}}_n(\mathcal{H}_{\mathrm{eff}})$ is strictly smaller than that of the static class, yielding a tighter generalization bound

$$\mathcal{E}_{\mathrm{gen}} \; \leq \; \hat{\mathfrak{R}}_n(\mathcal{H}_{\mathrm{eff}}) \; + \; \mathcal{O}\Big(n^{-1/2}\Big), \tag{19}$$

and, under a PAC-Bayes view, a smaller posterior KL term due to better data-congruent conditioning.

Data-dependent computation prunes portions of the hypothesis space on many inputs. Local Rademacher complexity bounds tighten when the effective class is smaller on typical samples. PAC-Bayes bounds also tighten when the posterior concentrates on input-conditioned submodels with reduced divergence from a data-informed prior.

## A.2 NETWORK CONFIGURATIONS

**Network architecture.** The point cloud sequence is first fed into the *Spatio-Temporal Embedding* module, where spatial convolution (*radius*, *nsamples*) and temporal convolution (*kernel size*, *stride*) are applied to map the input into spatio-temporal feature representations. Next, the features are passed into the *DST-Adapter*, which consists of a *SpatialAdapter* (token mixing + channel LoRA) and a *TemporalAdapter* (depthwise 1D convolution). Together with the global spatio-temporal statistics extracted by the textitContextEncoder, the adapter employs gating weights $(\alpha, \beta, \gamma)$ generated via **SiLU** activation to dynamically fuse spatial, temporal, and baseline features, followed by *FiLM modulation* for adaptive scaling and shifting. Finally, the enhanced features are fed into the *Transformer*, where each layer is composed of an *Attention* and a *FeedForward* block. The *Attention* block includes LayerNorm, QKV projection, multi-head attention, Softmax distribution, and spatial bias, with residual connections for stability. The *FeedForward* block consists of LayerNorm, two fully connected layers, GELU, and Dropout, also with residual connections. By stacking multiple such layers, the model produces globally modeled spatio-temporal representations.

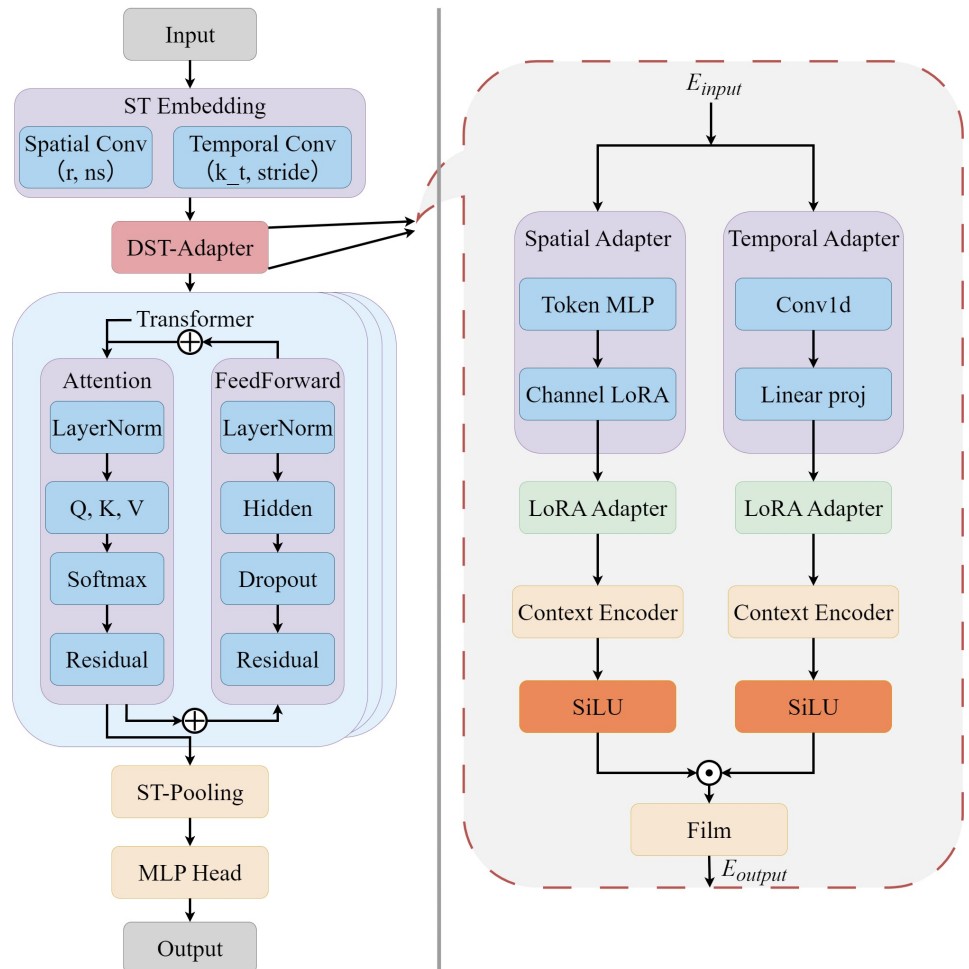

Figure 5: Structure of DSTA4D. The left side shows the overall process, and the right side shows the specific process of DST-Adapter.

## A.3 Discussion And Implications

Theorems A.1–A.1 justify the risk advantage of conditional mixtures over static pipelines and show that DST-Adapter approaches the per-sample best-expert risk when the gating is reasonably accurate. Theorem A.1 formalizes the benefit of performing spatio-temporal decoupling at the embedding layer: it suppresses cross-term error propagation early and enables variance reduction through content-aware gating. Theorem A.1 shows the principled, budget-aware compute allocation implemented by the gating, while Theorem A.1 explains FiLM's role in expanding conditional expressivity without destabilizing training. Finally, Theorem A.1 provides a generalization perspective: data-dependent compute shrinks the effective hypothesis complexity, tightening the generalization bound.

Together, these results provide a theoretically grounded rationale for the pre-ST decoupling and DST-Adapter design: they yield not only better expressivity-to-stability trade-offs and risk guarantees but also compute-optimal, content-aware adaptation on heterogeneous 4D point cloud videos.

## A.4 Additional baselines

**3D Action Recognition.** In order to provide a more comprehensive comparison, we further report quantitative results of additional baselines. Table 6 is presented as a continuation of Tab. 1. As shown in Tab. 6, early depth-based methods achieve below 82%, skeleton-based Actionlet reaches 88.21%, and point-based methods vary widely (PointNet++ only 61.61%, MeteorNet up to 88.50%).

In contrast, our approach achieves a new state-of-the-art accuracy of **96.17%**, clearly demonstrating the effectiveness of our spatio-temporal modeling strategy.

Table 6: Action recognition accuracy comparison on a standard benchmark.

| Methods | Input | Accuracy (%) |
|---|---|---|
| Vieira et al.(Vieira et al., 2012) | depth | 78.20 |
| Kläser et al.(Marszalek, 2008) | depth | 81.43 |
| Actionlet(Wang et al., 2012) | skeleton | 88.21 |
| PointNet++(Qi et al., 2017b) | point | 61.61 |
| MeteorNet(Liu et al., 2019) | point | 88.50 |
| DSTA4D(Ours) | point | **96.17** |

**Action segmentation on HOI4D.** We conduct the evaluation on the HOI4D dataset for the action segmentation task. Following the official split, the dataset comprises 2,971 scenes for training and 892 scenes for testing, where each sequence consists of 150 frames with 2,048 points per frame. Table 7 presents the quantitative comparison results on the HOI4D action segmentation task. We compare our proposed method against several state-of-the-art baselines, including P4Transformer(Fan et al., 2021a), PPTr(Wen et al., 2022), and the strong baseline X4D-SceneFormer(Jing et al., 2024). Consistent with the standard evaluation protocol, we report the frame-level accuracy (Acc), segmental edit distance (Edit), and segmental F1 scores at overlapping thresholds of 10%, 25%, and 50%. As shown in the table, our method consistently outperforms all competing approaches across all metrics. Specifically, built upon the X4D-SceneFormer backbone, our approach improves the frame-level accuracy from 84.1% to 85.4% and achieves the highest F1 scores, demonstrating the effectiveness of our design in capturing fine-grained temporal dynamics in 4D point cloud videos.

Table 7: Quantitative comparison of action segmentation results on the HOI4D dataset.

| Method | Acc | Edit | F1@10 | F1@25 | F1@50 |
|---|---|---|---|---|---|
| P4Transformer(Fan et al., 2021a) | 71.2 | 73.1 | 73.8 | 69.2 | 58.2 |
| P4Transformer + C2P(Zhang et al., 2023) | 73.5 | 76.8 | 77.2 | 72.9 | 62.4 |
| PPTr(Wen et al., 2022) | 77.4 | 80.1 | 81.7 | 78.5 | 69.5 |
| PPTr + C2P(Zhang et al., 2023) | 81.1 | 84.0 | 85.4 | 82.5 | 74.1 |
| PvNeXt(Wang et al., 2025) | 78.5 | 84.5 | 85.6 | 82.4 | 73.0 |
| X4D-SceneFormer(Jing et al., 2024) | 84.1 | 91.1 | 92.5 | 90.8 | 84.8 |
| DSTA4D(Ours) | **85.4** | **91.5** | **92.9** | **91.5** | **85.5** |

## A.5 ADDITIONAL QUALITATIVE RESULTS

**Qualitative results.** We also visualize several attention weights of the Transformer in Fig. 6. For the input, color encodes depth; in the second row of the attention visualization, colors closer to the top of the color bar indicate higher weights. As expected, the Transformer is able to accurately focus on the relevant regions across different frames. Furthermore, in the third row of the attention results, red highlights the attended regions while gray denotes the unattended areas. The results demonstrate that our attention mechanism not only captures the locations of actions but also emphasizes the critical transitional parts that connect different motions. This observation supports our intuition that the Transformer can serve as a substitute for explicit point tracking when modeling the spatio-temporal structure of point cloud videos.

**4D Semantic segmentation.** Figure 8 presents additional qualitative results for 4D semantic segmentation. The first column shows the raw input color images, the second column provides the ground-truth (GT) annotations, and the third column illustrates our predictions. As observed, even under complex environments and challenging scenarios, our approach consistently produces accurate segmentation maps that effectively separate semantic categories. These visual results further highlight the robustness and strong generalization capability of our method across diverse settings.

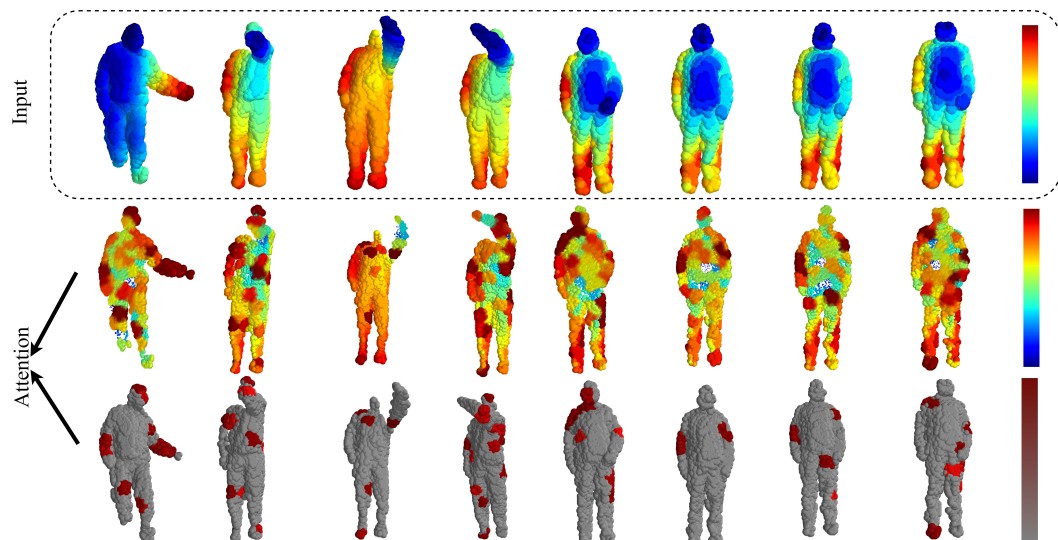

Figure 6: 3D Qualitative Results. The rows from top to bottom show the original input, the colored attention visualization, and the high-contrast attention visualization.

### A.6 GATING WEIGHTS RESULTS

We conducted a statistical analysis of the gating weights during training. As illustrated in Fig. 7, the trajectories differ markedly across datasets, confirming the mechanism's *content-aware* capability. On **MSR-Action3D**, the spatial weight ($\beta$) steadily increases and eventually dominates, while both the temporal weight ($\gamma$) and identity branch ($\alpha$) gradually decline. This indicates that the model quickly identifies spatial geometry as the primary discriminative cue and maintains this focus throughout training. In contrast, on **NTU RGB+D**, we observe a pronounced *"temporal surge"* in the mid-training phase (Epochs 5–8), where the temporal weight ($\gamma$) peaks around 0.50. Subsequently, the emphasis shifts toward the spatial branch ($\beta \to 0.567$), while the identity weight drops sharply ($\alpha < 0.08$). These findings demonstrate that the gating weights are not static parameters but evolve dynamically in response to training progress and dataset characteristics. The mechanism effectively acts as a *selector*, emphasizing temporal branches in motion-intensive scenarios and spatial branches in geometry-dominant contexts.

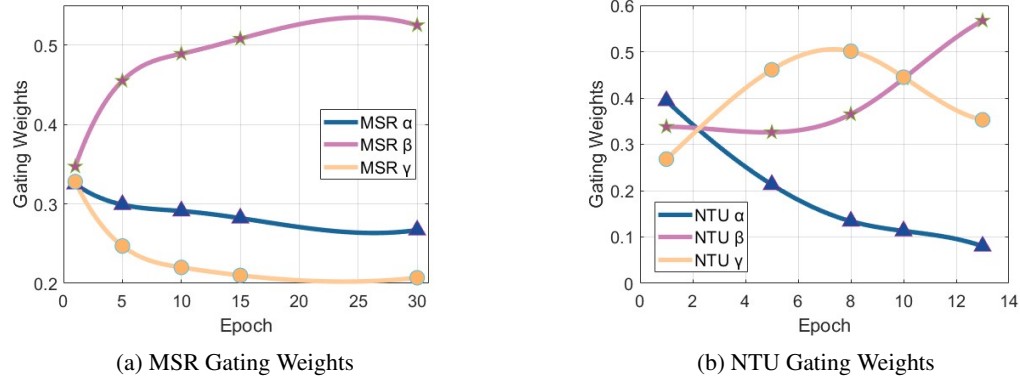

(a) MSR Gating Weights

(b) NTU Gating Weights

Figure 7: Evolution of gating weights on MSR-Action3D and NTU RGB+D.

## A.7 USAGE OF LLMS

We utilized a Large Language Model (LLM) to aid in polishing and refining the language in this paper. All core ideas, experimental designs, and analyses were conceived and executed by the authors. The LLM was used solely to improve the clarity and fluency of the text.

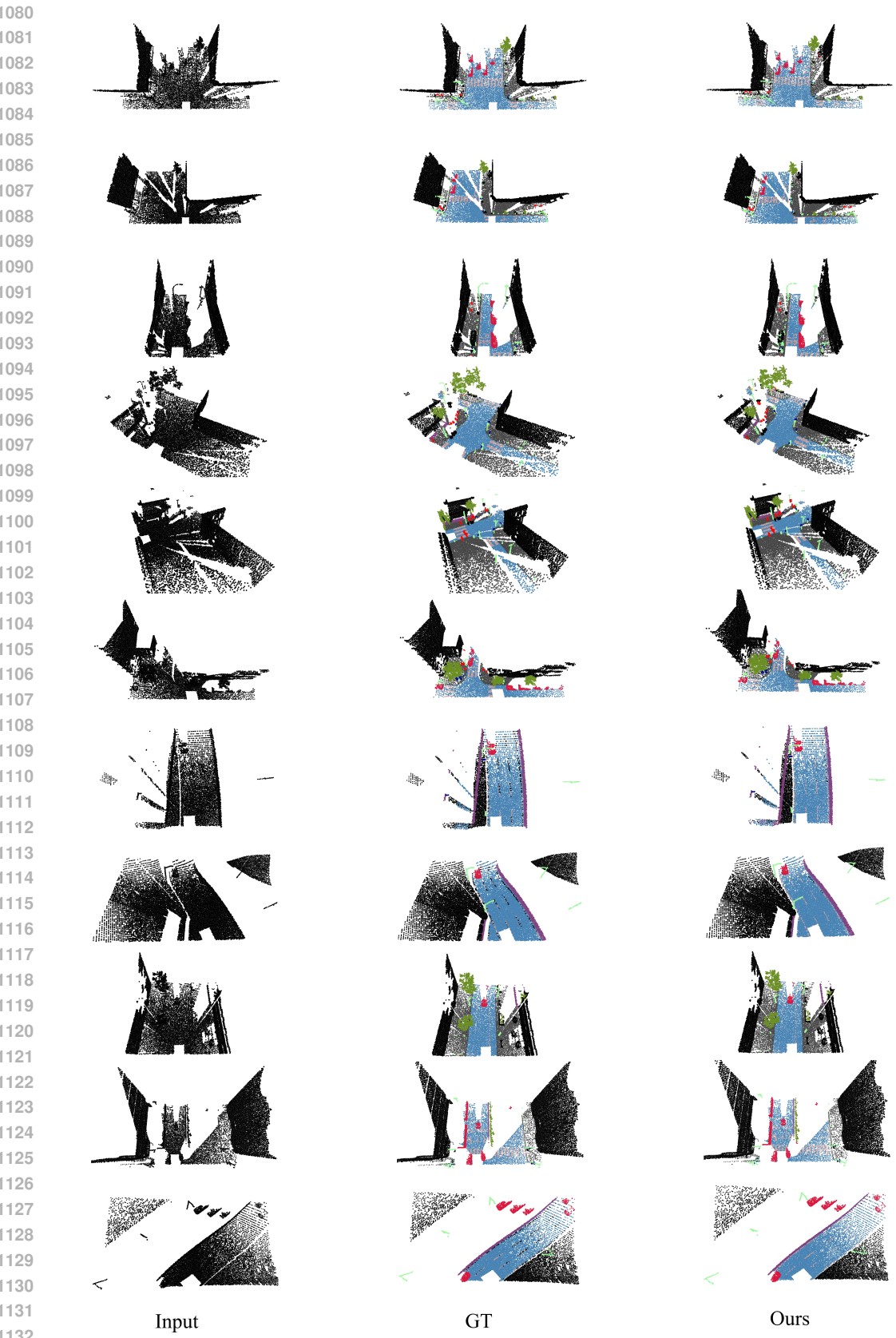

Input                          GT                          Ours

Figure 8: Qualitative results of Syn4D.

