# OpenReview forum: "DSTA4D: Rethinking Adaptive Spatio-Temporal Decoupling for Dynamic Point Cloud Videos"
_ICLR.cc/2026/Conference — Submitted to ICLR 2026_

### Official Review · Reviewer_cGvm · 2025-10-22

**Soundness:** 3
**Presentation:** 2
**Contribution:** 2
**Rating:** 4
**Confidence:** 5

**Summary:**

This paper proposes a novel light-weight spatial temporal decoupling strategy with respective adaptation. Three parallel paths including original path, spatial-enhancement path, and temporal enhancement one are designed. Gating mechanism and FiLM-based methods are also introduced. Even though this method achieves state-of-the-art performance on 4D datasets, I doubt the novelty, overclaim advantages, and efficiency.

**Strengths:**

1.The proposed module is lightweight, and is effective on 4D tasks.

2.Authors leverage several detailed designs in spatial enhancement, temporal enhancement, and gating to provide better spatio-temporal modeling capabilities.

3.Strong performance: The method achieves state-of-the-art performance on three commonly-used datasets like MSRA3D, NTU, and Synthia4D.

**Weaknesses:**

1.Limited Technique Novelty: Even though authors propose a lightweight but effective method with higher performance, many designs resort to existing techniques like token-mixing, channel-mixing, gating, and scaling&shifting. These incremental contributions have limited influence on this field. Furthermore, this pipeline retains the original point 4D conv and transformer backbones, which only add the designed components in between.

2.Overclaim: Authors claim ‘DST-Adapter o intelligently allocate computational resources, prioritizing the most critical spatio-temporal features’. However, in both experimental and method sections, there are no corresponding theory analysis or experimental proof. More explanations and experiments should be added to prove this point.

3.Similarly, authors say ‘Fixed computational graphs and processing strategies in Mamba-based methods can become a performance bottleneck with large-scale, high-frame-rate inputs.’ in the introduction section. But there are no corresponding analysis in the experiment section either.

4.Inconsistent Contents in Pipeline Figure. In the pipeline, what is the meaning of ‘symmetric’ part? There are no corresponding descriptions in the text. Also, it is quite confusing why the outputs of context encoder are $\alpha$, $\beta$, etc. In the text description, the output is ctx.

5.Efficiency: I have concerns on the additional computational resources since the methods keep the original 4D conv and transformer backbones. More analysis on how much additional overhead this method will incur should be added. Also, how authors balance the trade-off between accuracy and efficiency?

6.Results on HOI4D dataset: HOI4D is a newly-proposed dataset, which is more commonly used for metric evaluation in recent works. Could you provide the results on this dataset?

7.Insufficient ablation studies: There are no ablation studies for content extractor part and gating, fusion module. I think these are also core novelties in this paper, which should be validated. Also, most hyper-parameter settings should be compared.

8.In Figure 2, could authors compare the qualitative results with other methods?

9.Inconsistent tenses: In the related work section, the first and third paragraph use the common tense while past tense is used in the second one.

10.Grammars: The overall grammar is great. But there are some missed space before (. For example, ‘Dynamic Spatio-Temporal Adapter(DST-Adapter)’ in the abstract.

**Questions:**

Please refer to the weakness part. Also, since authors claim this adaptive decoupling is a universal design, could you provide more plug-and-play results on other baselines?

---

> ### Author Response · Authors · 2025-11-21
> **Response to Reviewer cGvm (Part 1)**
>
> > **W.1: Incremental contributions have limited influence.**
>
> We sincerely appreciate the reviewer’s conscientious assessment regarding the novelty of our method. We fully acknowledge the concern that our approach may appear to "borrow existing techniques" or "retain the original backbone structure." In this revision, we aim to articulate the design motivation and core contributions of **DSTA4D** more explicitly. We wish to emphasize that our work is not a mere combination of existing modules, but rather a novel, **context-driven three-path dynamic decoupling paradigm** specifically tailored to the characteristics of 4D point cloud videos.
>
> **1. Motivation: Addressing Limitations in Fixed Decoupling**
> Existing 4D point cloud modeling approaches predominantly adopt a fixed "Spatial-Temporal" two-stream decoupling strategy. However, we argue that this rigid paradigm lacks the adaptive capacity to handle videos with varying dynamic intensities and geometric complexities. Specifically, it tends to introduce noise in low-dynamic scenes and fails to extract sufficient features in high-dynamic scenarios, thereby imposing significant pressure on subsequent Transformers for long-term modeling. Addressing these under-discussed issues—specifically by introducing context-driven "three-path dynamic decoupling" at the embedding stage—is one of the key insights of this paper.
>
> **2. Systemic Structural Design**
> Our design represents a systemic structural innovation rather than a mechanical insertion of components into the backbone. It is grounded in three strategic considerations:
>
> * **(1) Introduction of an Identity Path:** This path preserves the stability of original features, ensuring the model does not introduce noise through excessive spatio-temporal modeling in static or weakly dynamic scenes.
> * **(2) Complementary Spatial/Temporal Enhancement:** The spatial path refines geometric semantics via token/channel mixing, while the temporal path utilizes lightweight convolutions to capture short-term dynamics, thereby enhancing the separability of action categories.
> * **(3) Synergy of Context Statistics, Gating, and FiLM:** By extracting global geometric and dynamic trends, this synergy achieves sample-level intelligent "computational allocation." This mechanism enables the model to aggressively activate temporal capabilities in high-dynamic scenes while prioritizing identity/stable features in low-dynamic ones, significantly improving the model's adaptability and discriminability.
>
> **3. Core Contribution**
> Holistically, the core contribution of DSTA4D lies not in any individual component (e.g., gating or scaling & shifting), but in pioneering the integration of **"context-driven three-path decoupling"** into 4D point cloud video modeling. This strategy aligns closely with the inherent sparsity, cross-frame inconsistency, and dynamic variability of point clouds. It effectively alleviates the burden on the backbone, allowing the Transformer to focus on long-range dependencies, thereby achieving significant improvements over SOTA while maintaining computational efficiency.
>
> We thank the reviewer for their valuable feedback. In the final version, we will further highlight these motivations, the logic behind our structural design, and our theoretical insights to ensure the method's novelty is clearly presented.

---

> ### Author Response · Authors · 2025-11-21
> **Response to Reviewer cGvm (Part 2)**
>
> > **W.2: Intelligently allocate computational resources.**
>
>
> We sincerely thank the reviewer for pointing out this critical gap. We acknowledge that our initial claim regarding "intelligent resource allocation" lacked sufficient empirical backing in the original manuscript. To address this, we have conducted a deep-dive analysis of the internal dynamics of the gating mechanism to validate how the model prioritizes spatio-temporal features.
>
> The proposed DST-Adapter employs a global context vector (`ctx`) and FiLM modulation to generate dynamic gating weights ($\alpha$: Identity, $\beta$: Spatial, $\gamma$: Temporal). These weights determine how much "modeling capacity" the network assigns to each branch. Following your suggestion, we tracked the evolution of these weights throughout the training process. As shown in **Table 1 and Table 2**, the model exhibits distinct, content-aware adaptation strategies for different datasets:
>
> 1.  **Stable Spatial Dominance on MSR-Action3D (Table 1):**
>     For the MSR dataset, we observe a monotonic increase in the spatial weight ($\beta$), rising from 0.347 to 0.525, while temporal ($\gamma$) and identity ($\alpha$) weights gradually decline. This suggests the model quickly identifies that spatial geometry is the primary discriminative cue for MSR samples and allocates its representational focus accordingly, rather than overfitting to temporal noise.
>
> 2.  **Dynamic Temporal "Surge" on NTU RGB+D (Table 2):**
>     In contrast, the NTU dataset triggers a more complex allocation strategy. We observe a significant **"temporal surge"** during the critical learning phase (Epochs 5–8), where the temporal weight ($\gamma$) peaks at $\approx 0.50$. This indicates the model intelligently prioritizes motion dynamics to resolve complex action ambiguities. Once temporal features are learned, the model shifts resources back to refining spatial details ($\beta \rightarrow 0.567$) and suppresses the identity connection ($\alpha < 0.08$), indicating a need for deep feature transformation rather than residual preservation.
>
> These results empirically prove that the gating mechanism is not static; it acts as an intelligent selector that dynamically reallocates feature importance based on data complexity. We have added the visualization curves, to **Appendix Fig. 7** of the revised manuscript.
>
> **Table 1: Evolution of Gating Weights on MSR-Action3D**
>
> | Epoch | $\alpha$ (Identity) | $\beta$ (Spatial) | $\gamma$ (Temporal) |
> | :---: | :---: | :---: | :---: |
> | **1** | 0.325 | 0.347 | 0.328 |
> | **5** | 0.299 | 0.455 | 0.247 |
> | **10** | 0.291 | 0.489 | 0.220 |
> | **15** | 0.282 | 0.508 | 0.210 |
> | **30** | 0.267 | 0.525 | 0.207 |
>
> **Table 2: Evolution of Gating Weights on NTU RGB+D**
>
> | Epoch | $\alpha$ (Identity) | $\beta$ (Spatial) | $\gamma$ (Temporal) |
> | :---: | :---: | :---: | :---: |
> | **1** | 0.394 | 0.338 | 0.268 |
> | **5** | 0.213 | 0.326 | 0.461 |
> | **8** | 0.134 | 0.365 | 0.501 |
> | **10** | 0.113 | 0.442 | 0.445 |
> | **13** | 0.080 | 0.567 | 0.353 |
>
>
>
> > **W.3: Similarly, authors say ‘Fixed computational graphs and processing strategies in Mamba-based methods can become a performance bottleneck with large-scale, high-frame-rate inputs.’ in the introduction section. But there are no corresponding analysis in the experiment section either.**
>
> We sincerely thank the reviewer for pointing out the lack of rigor in our Introduction. We sincerely thank the reviewer for pointing out the lack of rigor in our Introduction. We acknowledge that our original description regarding “fixed computational graphs” was not sufficiently precise. It may have unintentionally been interpreted as questioning Mamba4D’s generalization capabilities or performance, which was never our intention.
>
> In fact, we regard Mamba4D as a highly valuable contribution to the field. By pioneering the integration of Selective State Space Models (SSMs) into 4D point cloud temporal modeling, it has opened new directions and provided significant inspiration. Mamba4D demonstrates unique strengths in spatiotemporal modeling—particularly due to the efficiency, parallelism, and theoretical foundations of SSMs in handling long sequences—and we hold this work in high esteem.
>
> Our original intent was simply to discuss general engineering challenges related to adaptivity that fixed paradigms may encounter in scenarios characterized by “high frame rates, long sequences, and sparse point clouds,” rather than to make any negative assessment of Mamba4D itself. Recognizing that our phrasing lacked supporting experiments in the initial submission and could be misleading, we have revised the manuscript to adopt more objective language. These changes have been **highlighted in the revised manuscript** to demonstrate our responsiveness to your constructive feedback.
>
> We once again thank the reviewer for the detailed comments, which have helped us significantly improve the rigor and clarity of our presentation.

---

> ### Author Response · Authors · 2025-11-21
> **Response to Reviewer cGvm (Part 3)**
>
> >**W.4: Pipeline inconsistency.**
>
> Following the reviewer's suggestion, we have revised the "Global Information Fusion and Refinement" section starting at line 195. Specifically, we have supplemented the original formulation to clearly articulate the relationship between $ctx$ and the parameters $\alpha, \beta, \gamma$. The updated equation is as follows:
>
> $[\alpha, \beta, \gamma] = \sigma\big(\text{MLP}_{\text{gate}}(ctx)\big) \in \mathbb{R}^{B \times 3}$
>
> These revisions are marked in red highlight in the manuscript.
>
> >**W.5: Efficiency concerns.**
>
> We sincerely thank the reviewer for raising this important concern. You noted that although the paper emphasizes “intelligent computation allocation” and “adaptive efficiency,” it does not report FLOPs, memory consumption, inference latency, or throughput. To address this gap, we conducted additional experiments under a unified hardware environment, comparing PSTTransformer and our proposed DSTA4D, with each metric measured three times to ensure statistical stability.
>
> The results in Table 3 show that DSTA4D incurs modest increases in parameters (+5.4%) and FLOPs (+3.1%), which stem from the introduction of the dynamic spatio-temporal adaptation module. However, in terms of inference efficiency, DSTA4D performs better: it reduces latency by about 3.2%, improves throughput by 3.5%, and correspondingly increases frame rate. These findings indicate that the adaptive module enhances representational capacity without introducing noticeable computational overhead, and even achieves slight efficiency gains in practice.
>
> Most notably, DSTA4D reduces GPU cached memory usage to 392 MB, representing a 28.7% reduction compared to PSTTransformer. This highlights the superior memory efficiency of DSTA4D, which makes it particularly suitable for scenarios involving multi-model parallelism or deployment under memory-constrained environments.
>
> In summary, although DSTA4D introduces a small increase in computational cost, it demonstrates clear advantages in latency, throughput, and memory efficiency. These results provide strong evidence that the proposed dynamic spatio-temporal adaptation module achieves a well-balanced trade-off between adaptability and practical efficiency. We have included these detailed efficiency measurements in the revised manuscript to comprehensively support our claims on “adaptive efficiency.”
>
>
> **Table 3. Efficiency Comparison between PSTTransformer and DSTA4D (3 runs, mean ± std, relative change vs. PST)**
>
> | Model                  | Params (M) ▼ | FLOPs (G) ▼ |     Latency (ms) ▼| Throughput (clips/s) ▲ |      Frames (fps) ▲ | GPU alloc (MB) ▼| GPU cached (MB) ▼|
> | ------------------ | ----------:| ---------:| ------------:| --------------------:| ------------:| --------------:| ---------------:|
> | **PSTTransformer** | 45.483     | 40.790    | 33.28 ± 0.14 | 30.05 ± 0.14         | 721.19 ± 3.0 | 182.22         | 550             |
> | **DSTA4D (Ours)**  | 47.942 (+5.40%) | 42.054 (+3.10%) | **32.21 ± 1.65 (−3.22%)** | **31.10 ± 1.45 (+3.50%)** | **746.45 ± 34.8 (+3.50%)** | 191.83 (+5.27%) | **392 (−28.73%)** |
>
>
> >**W.6: Add HOI4D results.**
>
> Thank you for your valuable suggestion. We agree that evaluating our method on the HOI4D dataset is important for demonstrating the generalizability of our approach. HOI4D is a newly-proposed large-scale 4D egocentric dataset focusing on category-level human-object interactions. Due to its diverse action categories, long sequences, and complex human-object interaction dynamics, HOI4D serves as a strong benchmark for fine-grained motion recognition.
>
> We have conducted experiments on the HOI4D action segmentation task. Our method achieves an **accuracy of 78.8%**, which surpasses the advanced method PPTr (77.4%) and PVNEXT(78.5%). The detailed comparison is shown below:
>
> **Table 4. Results of HOI4D**
> | Method        | Length | Accuracy |
> |---------------|--------|----------|
> | P4Transformer | 150    | 71.2     |
> | P4 + adapter  | 150    | 75.5     |
> | PPTr          | 150    | 77.4     |
> |  PVNEXT       | 150    | 78.5     |
> |  **Ours**     | 150    | 78.8     |

---

> ### Author Response · Authors · 2025-11-21
> **Response to Reviewer cGvm (Part 4)**
>
> >**W.7: Insufficient ablation studies.**
>
> We sincerely thank the reviewer for this constructive suggestion. We agree that validating the contribution of each core component and analyzing hyper-parameter sensitivity is crucial for demonstrating the robustness of our proposed method. In the revised manuscript, we have conducted comprehensive ablation studies to address these concerns.
>
> **1. Effectiveness of Decoupling Branches and Gating Mechanism**
> To validate the specific designs of our **DSTA4d** model, particularly the decoupled branches and the fusion module, we performed extensive experiments. The results are summarized in **Table 5** and **Table 6** below.
>
> * **Impact of Decoupled Branches:** We evaluated the model by removing the Identity, Temporal, and Spatial branches individually. As shown in **Table 5**, removing any branch leads to a performance drop, confirming their complementary nature. Specifically, the Spatial branch proves to be the most critical component (acc. drops to 93.72%). Furthermore, our full model significantly outperforms the "Normal decoupling" baseline (using simple concatenation), verifying the effectiveness of our sophisticated decoupling design.
>
> * **Comparison of Fusion Strategies:** We compared our proposed **Gating** mechanism against standard fusion methods: Summation (`Sum`) and Concatenation (`Concat`). As shown in **Table 6**, the adaptive Gating mechanism achieves the best accuracy (**96.17%**), surpassing `Sum` by 1.05% and `Concat` by 2.80%. This validates our hypothesis that dynamically adjusting the fusion weights ($\alpha, \beta, \gamma$) based on context is superior to static aggregation.
>
> **Table 5: Ablation study on decoupling branches.**
> | Model Setting | Accuracy (%) | $\Delta$ |
> | :--- | :---: | :---: |
> | **Our Method (Full Model)** | **96.17** | **-** |
> | w/o Identity branch | 95.47 | -0.70 |
> | w/o Temporal branch | 94.42 | -1.75 |
> | w/o Spatial branch | 93.72 | -2.45 |
> | Normal decoupling (Baseline)| 93.37 | -2.80 |
>
> **Table 6: Comparison of different fusion strategies.**
> | Fusion Strategy | Accuracy (%) |
> | :--- | :---: |
> | Concat (Concatenation + Conv) | 93.37 |
> | Sum (Element-wise Summation)| 95.12 |
> | **Gating (Ours)** | **96.17** |
>
> **2. Hyper-parameter Sensitivity Analysis**
> We also investigated the impact of the Context Vector Dimension (`ctx_dim`), which is a key hyper-parameter in our context encoder. As illustrated in **Table 7**, the model performance initially improves as the dimension increases, peaking at **128**. Further increasing the dimension (to 256 or 512) leads to a slight performance degradation, likely due to feature redundancy or overfitting. This analysis confirms that our default setting of 128 provides the optimal balance between representation capacity and model complexity.
>
> **Table 7: Impact of Context Vector Dimension (`ctx_dim`).**
> | `ctx_dim` | 64 | 96 | **128 (Default)** | 256 | 512 |
> | :--- | :---: | :---: | :---: | :---: | :---: |
> | Accuracy (%)| 94.07 | 95.12 | **96.17** | 94.07 | 93.73 |
>
> We have integrated these analyses and tables into Appendix of the revised paper.

---

> ### Author Response · Authors · 2025-11-21
> **Response to Reviewer cGvm (Part 5)**
>
> >**W.8: No qualitative comparison.**
>
> We thank the reviewer for the constructive suggestion. To improve the visual interpretability of the qualitative analysis presented in Figure 2, we have provided supplementary visualization comparisons against other baselines in Figure 4.
>
> >**W.9: Inconsistent tenses.**
>
> We sincerely thank the reviewer for the careful review and valuable suggestions regarding the writing details of our manuscript. Addressing the issue of inconsistent tenses in the "Related Work" section, we have conducted a thorough check of the full text. We have unified the tense in the second paragraph by changing the original past tense to the present tense to align with the surrounding context, thereby ensuring logical consistency and reading fluency. The modified content is clearly marked in red highlight in the revised manuscript (Revision location: Lines 117–127). We thank the reviewer again for this reminder, which has helped make our presentation more rigorous.
>
> >**W.10: Grammars.**
>
> We sincerely thank the reviewer for the meticulous corrections regarding the abstract and grammatical details throughout the paper. Regarding the missing space in "Dynamic Spatio-Temporal Adapter(DST-Adapter)" in the abstract, we have corrected this in the revised version. Furthermore, we have conducted a line-by-line proofreading of the full text covering grammar, punctuation, and formatting to ensure no similar oversights remain. All relevant revisions are marked in red highlight in the revised manuscript (Revision location: Abstract, lines 21–23). We highly value your suggestions, which have significantly contributed to improving the overall presentation quality of our work.
>
> >**Q.1: More plug-and-play results on other baselines.**
>
> We sincerely thank the reviewer for this insightful suggestion regarding the universality of our design. To verify the "plug-and-play" capability and generalization of our proposed GSTA4d (DSTA) module, we integrated it into two representative point cloud video modeling backbones: **P4Transformer** and **PST-Transformer**. We conducted ablation studies to measure the performance gains on these baselines.
>
> As shown in **Table 8**, our module consistently improves the performance of different architectures:
> * **On P4Transformer:** The integration of our module yields an accuracy of **93.03%**, achieving a significant improvement of **+2.09%** over the baseline (90.94%).
> * **On PST-Transformer:** Our method boosts the accuracy to **96.17%**, resulting in a **+2.44%** performance gain compared to the original model (93.73%).
>
> **Table 8: Plug-and-play performance comparison on P4Transformer and PST-Transformer backbones.**
>
> | Method | Accuracy (%) |
> | :--- | :--- |
> | P4Transformer | 90.94 |
> | P4Transformer + DSTA | **93.03** |
> | PST-Transformer | 93.73 |
> | PST-Transformer + DSTA | **96.17** |
>
>
> These results empirically demonstrate that our adaptive decoupling design is robust and effective across different backbones, supporting its potential as a universal module for point cloud video understanding.

---

> ### Comment · Reviewer_cGvm · 2025-11-27
>
> Thanks for the authors' careful consideration and additional experiments related to our concerns. This feedback addressed well about the claim, ablation studies, and efficiency. However, I still have some concerns；
>
> 1. For the allocation of computational resources, what causes the differences between different datasets? i.e., Why on MSRA, it has the spatial dominance, while on NTU, it first prioritizes spatial then temporal ones. Also, I question the necessity of this allocation: for the previous pipeline they also mostly decouple spatial and temporal modelings, how these works allocate resources? I am curious about whether it is the proposed method that influences the allocation, or it is also the same to other works.
>
> 2. Even though authors add experiment results, they are still worse than recent SOTA methods, like C2P, X4D-SceneFormer, etc.
>
> 3. I still think the novelty is somewhat limited, which is built on some current exisiting pipelines.

---

> > ### Author Response · Authors · 2025-11-28
> > **Response to Reviewer cGvm**
> >
> > > **Q1: Analysis of Resource Allocation Mechanism**
> >
> > We appreciate the reviewer's insightful query regarding the underlying causes and necessity of our resource allocation strategy. We clarify that the observed allocation patterns are data-driven results enabled by our dynamic architecture, distinct from the fixed designs in previous works.
> >
> > Cause of Differences: Intrinsic Data Physics The difference in allocation strategies between MSR and NTU is not a random artifact but a reflection of their distinct physical characteristics:
> >
> > * MSR: This task relies heavily on fine-grained geometry to distinguish categories, while the temporal movement is often simple or transitional. Consequently, our model realizes this redundancy and aggressively allocates resources to the Spatial Path to capture subtle geometric details, suppressing the Temporal path to avoid noise.
> >
> > * NTU: This dataset involves complex whole-body dynamics. The model exhibits a Temporal Surge (prioritizing spatial first to identify the subject, then significantly increasing temporal weights) because capturing long-range motion dependencies is critical for resolving action ambiguity in this context.
> >
> > Necessity and Differentiation from Previous Works
> >
> > * Previous Works (Static Allocation): Most existing decoupled methods utilize a pre-defined ratio of spatial-to-temporal layers (e.g., strictly alternating blocks) regardless of the input data type. They do not allocate resources. This leads to inefficiency: wasting computation on temporal modeling for static-dominant samples or under-fitting rapid dynamics in complex actions.
> >
> > * Our Method (Dynamic Allocation): The necessity of our method lies in its ability to break this rigidity. While previous works force every sample through the same pipeline, our Context-Driven Gating acts as an intelligent controller that perceives the data's complexity and actively reconfigures the computing budget.
> >
> > It is indeed our proposed method that enables the allocation capability, but it is the dataset properties that determine the specific allocation strategy. This adaptability is exactly what static backbones lack.
> >
> > > **Q2: Positioning DSTA4D within the SOTA Landscape**
> >
> > We sincerely thank the reviewer for highlighting these advanced SOTA methods. We fully agree that X4D-SceneFormer and C2P  are pioneering works that have significantly pushed the boundaries of 4D point cloud understanding through Cross-Modal Knowledge Transfer and Self-Supervised Pre-training, respectively.
> >
> > While our absolute accuracy is lower than these specific methods, we respectfully suggest that this performance gap stems primarily from distinct experimental settings and resource utilization, rather than architectural limitations. To provide a comprehensive evaluation, we clarify the positioning of DSTA4D as follows:
> >
> > * Setting Discrepancy: Multi-Modal vs. Single-Modal.X4D-SceneFormer achieves exceptional results by leveraging RGB sequences as a privileged modality during training to transfer texture priors. This "Multi-Modal Training" paradigm provides rich semantic cues that are unavailable in pure geometric data. Our Approach: DSTA4D operates in a strict Single-Modal (Geometry-only) setting. We focus on maximizing the representational power of raw point clouds without relying on external visual data. In this Geometry-only track, our method outperforms other single-modal baselines (e.g., PPTr, PVNEXT), demonstrating the efficacy of our dynamic allocation mechanism in mining intrinsic geometric features.
> >
> > * Training Paradigm: Pre-training vs. From Scratch. C2P utilizes a robust Self-Supervised Pre-training framework, which requires a computationally intensive two-stage process on large-scale unlabeled data to initialize weights. Our Approach: DSTA4D is trained From Scratch. We demonstrate that even without the benefit of large-scale pre-training, optimizing the internal computational flow allows for highly competitive performance.
> >
> > * Compatibility and Orthogonality Crucially, our contribution is architectural and orthogonal to these methods. The DSTA module is a plug-and-play dynamic encoder that could theoretically be integrated into the point-branch of X4D or combined with C2P’s pre-training strategy to unlock further gains. While adapting our method to these complex multi-modal/pre-training frameworks was not feasible within the rebuttal timeframe, our results on standard backbones (P4Transformer, PST-Transformer) and three datasets(MSR,NTU,SYN) confirm that DSTA4D is a universal performance booster.
> >
> > In summary, DSTA4D achieves SOTA performance within the Single-Modal, From-Scratch setting. We believe this work offers a valuable, efficient architectural distinct from heavy multi-modal pipelines. We genuinely hope this clarification highlights the distinct value of our contribution and would be grateful if you could consider a positive reassessment of our work.

---

> > ### Author Response · Authors · 2025-11-28
> > **Response to Reviewer cGvm**
> >
> > > **Q3: Clarification on Novelty and Contributions**
> >
> > We strictly acknowledge the reviewer’s comment that our method is implemented upon established backbones. However, we respectfully argue that our contribution is not an incremental addition of modules, but a paradigm shift from Static Architecture to Dynamic Computation Allocation.
> >
> > * Identifying the Fundamental Flaw in Existing Pipelines: Most existing pipelines adopt a "One-Size-Fits-All" strategy (e.g., fixed Spatial-Temporal layers) for all inputs. Our research identifies that this rigid design is fundamentally inefficient for 4D point clouds, which exhibit extreme variance in information density (e.g., subtle finger movements vs. whole-body running). Existing pipelines fail to adapt to these fluctuations, leading to either redundancy or under-fitting.
> >
> > * From Component Stacking to Intelligent Control: The novelty of DSTA4D lies in its Context-Driven Dynamic Decoupling mechanism. Unlike standard modules that passively process features, our design acts as an active controller. It perceives the global context of each sample and dynamically reconfigures the importance of Identity, Spatial, and Temporal paths. This transforms the backbone from a static feature extractor into an adaptive system that knows whether to focus on geometry or motion.
> >
> > * Universal Methodology over Specific Architecture: By building on existing pipelines, we actually demonstrate the universality of our insight. The proposed dynamic mechanism addresses the inherent static bottleneck of current deep learning models. As shown in our plug-and-play experiments, this design philosophy can revitalize various existing architectures, offering a generalizable direction for future efficient 4D modeling.
> >
> > In summary, our novelty lies in redefining how computational resources are allocated within a pipeline, rather than simply creating a new pipeline structure.

---

> > ### Author Response · Authors · 2025-12-03
> > **Response to Reviewer cGvm**
> >
> > > Q2: Positioning DSTA4D within the SOTA Landscape (additional)
> >
> > To further address the performance concerns raised, we carried out additional experiments on the HOI4D dataset during the rebuttal phase. Specifically, we adopted the training strategy of X4D-SceneFormer. As noted in recent literature, X4D-SceneFormer achieves exceptional results by leveraging RGB sequences as a privileged modality during training to transfer texture priors. This "Multi-Modal Training" paradigm provides rich semantic cues that are unavailable in pure geometric data. By incorporating this experimental setting, our method successfully integrates these semantic cues and achieves state-of-the-art performance, surpassing previous methods including X4D-SceneFormer with an accuracy of 85.4% and an Edit score of 91.5%, as detailed in the updated table.
> >
> >
> > | Method | Acc | Edit | F1@10 | F1@25 | F1@50 |
> > | :--- | :---: | :---: | :---: | :---: | :---: |
> > | P4Transformer | 71.2 | 73.1 | 73.8 | 69.2 | 58.2 |
> > | P4Transformer + C2P | 73.5 | 76.8 | 77.2 | 72.9 | 62.4 |
> > | PPTr | 77.4 | 80.1 | 81.7 | 78.5 | 69.5 |
> > | PPTr + C2P | 81.1 | 84.0 | 85.4 | 82.5 | 74.1 |
> > | X4D-SceneFormer | 84.1 | 91.1 | 92.5 | 90.8 | 84.8 |
> > | **Ours** | **85.4** | **91.5** | **92.9** | **91.5** | **85.5** |

---

### Official Review · Reviewer_GSr5 · 2025-10-27

**Soundness:** 3
**Presentation:** 3
**Contribution:** 3
**Rating:** 6
**Confidence:** 4

**Summary:**

The paper addresses dynamic 4D point cloud video understanding, noting that existing models use static, uniform pipelines that poorly handle varying spatio-temporal complexity. It proposes DSTA4D, a content-aware architecture that first decouples spatial and temporal features at the embedding layer, then applies a Dynamic Spatio-Temporal Adapter (DST-Adapter). The DST-Adapter uses a global context encoder to generate FiLM-style gating weights that fuse three parallel feature streams (identity, spatial-enhanced, temporal-enhanced) in a data-dependent way. This mechanism adapts the computational focus to the scene complexity. Experiments on MSR-Action3D, NTU RGB+D, and Synthia4D show notable gains: +5.23% action accuracy on MSR-Action3D, +1.0% on NTU RGB+D, and +1.36% mIoU on Synthia4D over previous state-of-the-art. These results establish new benchmarks, validating the approach’s effectiveness on classification and segmentation tasks.

**Strengths:**

- Strong empirical gains. The method achieves state-of-the-art performance on all evaluated tasks. For example, it sets a new record on MSR-Action3D (96.17% vs. ~94.8% prior), and improves segmentation mIoU on Synthia4D by ~0.8 points. These consistent improvements on diverse benchmarks indicate the approach’s effectiveness.

- Rigorous evaluation. The experiments are comprehensive: multiple datasets (small to large, action recognition and segmentation), standard splits, and many baselines are compared. Ablations in Table 4 show each component’s importance (removing FiLM or decoupling hurts accuracy), giving confidence in the design choices. The paper also analyzes hyperparameters (e.g., temporal kernel size) and compares to baselines using more frames.

- Theoretical backing. The inclusion of theoretical results (Appendix Theorem A.1) provides a solid foundation, explaining why decoupling and conditional gating can improve risk bounds. This adds depth and credibility beyond empirical claims.

**Weaknesses:**

- No efficiency metrics: Along similar lines, the paper lacks any evaluation of resource usage (GPU time, memory, etc.). It would strengthen the work to know if the adapter adds significant overhead compared to a baseline Transformer. Without this, the practicality of “lightweight” gating is speculative.

- Limited long-range modeling: The temporal adapter is a depthwise conv with kernel size 3, which captures only very short-term motion. Although the paper claims “long-term dynamic perception” via decoupling, it is actually the downstream Transformer layers that must capture longer dependencies. The text could better clarify this interaction. As is, one might question whether decoupling truly addresses long-term dynamics or mainly local enhancements.

- Dataset scope: The evaluation, while diverse, is still limited to a small action dataset (MSR-Action3D), a large human-action set (NTU), and one synthetic driving dataset. No real-world outdoor or multi-object benchmarks (e.g. SemanticKITTI sequence modeling) are tested. It would be useful to see how DSTA4D generalizes to more complex or varied point-cloud video domains.

**Questions:**

- Effect of gating on compute: Since the gating weights merely re-weight the three branches, have the authors considered sparsifying them (e.g. setting small weights to zero) to actually skip computation? Can you clarify what resource savings (if any) are achieved by the adapter? Is there a runtime or FLOP count comparison to a baseline Transformer?

- Temporal modeling scope: The Temporal Adapter uses a small 1D conv (kernel size 3). Do you rely on the subsequent Transformer layers for longer-range temporal dependencies? Would it help to use a larger temporal window or learnable RNN/SSM in that branch? What was the reasoning for this design?

- Context vector details: What is the dimensionality of the context vector $ctx$? Did you experiment with different sizes or pooling strategies? How sensitive is performance to this choice, or to the structure of the gating MLP?

---

> ### Author Response · Authors · 2025-11-21
> **Response to Reviewer GSr5 (Part 1)**
>
> > **W.1: No efficiency metrics.**
>
> We sincerely thank the reviewer for raising this important concern. You noted that although the paper emphasizes “intelligent computation allocation” and “adaptive efficiency,” it does not report FLOPs, memory consumption, inference latency, or throughput. To address this gap, we conducted additional experiments under a unified hardware environment, comparing PSTTransformer and our proposed DSTA4D, with each metric measured three times to ensure statistical stability.
>
> The results in Table 1 show that DSTA4D incurs modest increases in parameters (+5.4%) and FLOPs (+3.1%), which stem from the introduction of the dynamic spatio-temporal adaptation module. However, in terms of inference efficiency, DSTA4D performs better: it reduces latency by about 3.2%, improves throughput by 3.5%, and correspondingly increases frame rate. These findings indicate that the adaptive module enhances representational capacity without introducing noticeable computational overhead, and even achieves slight efficiency gains in practice.
>
> Most notably, DSTA4D reduces GPU cached memory usage to 392 MB, representing a 28.7% reduction compared to PSTTransformer. This highlights the superior memory efficiency of DSTA4D, which makes it particularly suitable for scenarios involving multi-model parallelism or deployment under memory-constrained environments.
>
> In summary, although DSTA4D introduces a small increase in computational cost, it demonstrates clear advantages in latency, throughput, and memory efficiency. These results provide strong evidence that the proposed dynamic spatio-temporal adaptation module achieves a well-balanced trade-off between adaptability and practical efficiency. We have included these detailed efficiency measurements in the revised manuscript to comprehensively support our claims on “adaptive efficiency.”
>
>
> **Table 1. Efficiency Comparison between PSTTransformer and DSTA4D (3 runs, mean ± std, relative change vs. PST)**
>
> | Model                  | Params (M) ▼ | FLOPs (G) ▼ |     Latency (ms) ▼| Throughput (clips/s) ▲ |      Frames (fps) ▲ | GPU alloc (MB) ▼| GPU cached (MB) ▼|
> | ------------------ | ----------:| ---------:| ------------:| --------------------:| ------------:| --------------:| ---------------:|
> | **PSTTransformer** | 45.483     | 40.790    | 33.28 ± 0.14 | 30.05 ± 0.14         | 721.19 ± 3.0 | 182.22         | 550             |
> | **DSTA4D (Ours)**  | 47.942 (+5.40%) | 42.054 (+3.10%) | **32.21 ± 1.65 (−3.22%)** | **31.10 ± 1.45 (+3.50%)** | **746.45 ± 34.8 (+3.50%)** | 191.83 (+5.27%) | **392 (−28.73%)** |

---

> ### Author Response · Authors · 2025-11-21
> **Response to Reviewer GSr5 (Part 2)**
>
> >**W.2: Limited long-range modeling.**
>
>
> We thank the reviewer for the insightful questions regarding the temporal modeling scope and the interaction between the Temporal Adapter and the Transformer. We appreciate the opportunity to clarify our design philosophy.
>
> **1. Reliance on Transformer for Long-Range Dependencies:**
> You are correct. We explicitly rely on the subsequent **Spatio-Temporal Transformer** layers to model **long-range dependencies**, while the **Temporal Adapter** ($k=3$) is designed to capture **short-term, local dynamics**.
> * **Hierarchical Design:** As described in Section 3.3, the Temporal Adapter utilizes a lightweight 1D convolution explicitly to capture *"local motion patterns and short-term trajectory information"*. Its goal is to extract precise, high-frequency motion cues (e.g., immediate frame-to-frame transitions) to form robust token representations.
> * **Global Context:** The subsequent Transformer then integrates these locally refined tokens to establish global temporal connections. This hierarchical "Local (Adapter) + Global (Transformer)" design allows each module to focus on what it does best.
>
> **2. Clarification on "Long-Term Perception via Decoupling":**
> We apologize if the phrasing was ambiguous. When we state that the model achieves *"long-term dynamic perception via decoupling"*, we mean that the decoupling strategy **facilitates** the Transformer's ability to model long sequences. By disentangling spatial and temporal features in the early embedding stage, we provide the Transformer with "cleaner" and distinct temporal tokens. This structural prior allows the self-attention mechanism to efficiently connect these cues across long horizons, rather than struggling to separate entangled spatio-temporal noise. Thus, decoupling is the *enabler* for effective long-term perception.
>
> **3. Reasoning for Small Kernel ($k=3$) vs. Larger Window:**
> We empirically investigated whether increasing the adapter's receptive field (via larger kernels) or using heavier modules would benefit the model. The results demonstrate that a smaller kernel is optimal:
>
> **Table 2. Ablation Results of Kernel Seize**
> | Temporal Kernel Size | Accuracy on MSR-Action3D |
> | :--- | :--- |
> | **3 (Ours)** | **96.17%** |
> | 5 | 95.47% |
> | 7 | 94.42% |
> | 9 | 93.73% |
>
> * **Performance Drop with Larger Scope:** As shown above, accuracy drops significantly as the kernel size increases to 9 ($93.73\%$). Larger kernels in the adapter branch tend to **over-smooth** the fine-grained local motion features, blurring the distinct signals that the Transformer needs.
>
> * **Efficiency vs. Redundancy:** Since the downstream Transformer is already a powerful global model, adding large-kernel Conv in the adapter introduces functional redundancy and computational overhead without performance gains. Our design ($k=3$) strikes the best balance between capturing sharp local dynamics and maintaining efficiency.
>
> **Table 3. Temporal Modeling Ablation**
> | Temporal Modeling | Accuracy on MSR-Action3D |
> | :--- | :--- |
> | **Ours** | **96.17%** |
> | SSM | 93.37% |
> | RNN | 93.73% |
>
> The ablation study results strongly validate the efficiency of the proposed lightweight design. "Ours" (utilizing a 1D Conv with $k=3$) achieved an accuracy of 96.17%, significantly outperforming both the Mamba-based (93.37%) and RNN-based (93.73%) variants.This outcome suggests that for the specific role of this adapter, local temporal continuity is far more critical than long-range dependency modeling. Complex sequence models like Mamba or RNNs likely introduced unnecessary optimization difficulty or redundancy, given that the backbone model may already handle global context. The simple 1D convolution provides a robust inductive bias for temporal smoothing, proving that a focused, lightweight operator is superior to heavier architectures for extracting high-frequency motion details in this setting.

---

> ### Author Response · Authors · 2025-11-21
> **Response to Reviewer GSr5 (Part 3)**
>
> >**W.3: Restricted dataset scope.**
>
> We sincerely appreciate the reviewer's constructive feedback regarding dataset diversity. We fully recognize the importance of evaluating our model on more complex and diverse point cloud video scenarios to ensure robust generalization.
>
> The three datasets currently utilized in the manuscript—MSR-Action3D (small-scale actions), NTU (large-scale multi-view human actions), and Syn—were selected to exhibit distinct variations in scene complexity, point cloud density, motion types, and temporal characteristics. These choices were intended to demonstrate the model's generalization capabilities across different task modalities. However, we acknowledge that these datasets predominantly focus on indoor or human-centric settings and do not fully cover large-scale outdoor environments characterized by multi-object, continuous temporal dynamics.
>
> We agree that SemanticKITTI represents a highly complex outdoor point cloud sequence, featuring large spatial ranges, long temporal spans, rich geometric structures, and multi-object dynamics. However, compared to action-centric datasets, SemanticKITTI differs significantly in data processing pipelines, label systems, and temporal features, necessitating substantial engineering adaptation. Following the experimental settings of widely cited baselines, SemanticKITTI has not typically been included in the main benchmarks for this specific task. As our current work focuses on establishing the core DSTA4D framework and completing primary benchmarks, we have not yet finalized experiments on SemanticKITTI. Nevertheless, we highly value this suggestion and plan to incorporate tests on such large-scale, real-world outdoor scenarios in future extended versions to further verify the model's adaptability and robustness.
>
> To strictly address the need for greater scenario complexity in this revision, we have introduced HOI4D, a rapidly emerging benchmark for point cloud Human-Object Interaction (HOI). HOI4D is characterized by high point density, rich interactive motions, diverse manipulated objects, and clear dynamic labels. Compared to pure action recognition datasets, HOI4D involves more complex human-object relationships, fine-grained geometric variations, and realistic operational scenarios. Consequently, it serves as a rigorous benchmark for evaluating spatiotemporal feature modeling and comprehensively demonstrates the advantages of our proposed Dynamic Spatio-Temporal Adapter (DSTA).
>
> We have conducted supplementary experiments on the HOI4D action segmentation task. As shown in Table 4, our method achieves an accuracy of 78.8%, surpassing advanced methods such as PPTr (77.4%) and PVNEXT (78.5%). These results have been incorporated into the revised manuscript to enhance the completeness of our experimental evaluation. We thank the reviewer again for this insightful advice, which has helped improve the applicability of our work across different scenarios.
>
> **Table 4. Results of HOI4D**
> | Method        | Length | Accuracy |
> |---------------|--------|----------|
> | P4Transformer | 150    | 71.2     |
> | P4 + adapter  | 150    | 75.5     |
> | PPTr          | 150    | 77.4     |
> |  PVNEXT       | 150    | 78.5     |
> |  **Ours**     | 150    | 78.8     |
>
> >**Q.1: Sparsification gating weights & resource savings**
>
> **Resource savings see W.1.**
>
> For Sparsification gating weights, we further conducted detailed ablations on the three decoupled branches (Identity, Spatial, Temporal) to verify their importance and complementarity. Results demonstrate that the full model (96.17%) consistently outperforms any variant missing a single branch:
> 1. Removing the Spatial branch → accuracy drops to 93.72%, highlighting the importance of spatial geometric interactions.
> 2. Removing the Temporal branch → accuracy drops to 94.42%, confirming the critical role of independent temporal modeling.
> 3. Removing the Identity branch → accuracy drops to 95.47%, showing that retaining the raw feature stream is necessary for information integrity and training stability.
>
> Compared to the baseline normal decoupling (93.37%), our multi-branch dynamic collaboration strategy achieves a clear performance gain. This demonstrates that the branches are not simply additive, but rather complementary, jointly enhancing the model’s ability to represent complex spatio-temporal data.
>
> **Table 5. Ablation Result of decoupling.**
> | Model Setting              | Accuracy (%) |
> | :------------------------- | -----------: |
> | **Our Method (Full Model)**| **96.17**    |
> | w/o Identity branch        | 95.47        |
> | w/o Temporal branch        | 94.42        |
> | w/o Spatial branch         | 93.72        |
> | Normal decoupling (Baseline)| 93.37       |

---

> ### Author Response · Authors · 2025-11-21
> **Response to Reviewer GSr5 (Part 4)**
>
> >**Q.2: Limited long-range temporal modeling scope, larger temporal window or learnable RNN/SSM and reasoning**
>
> Refer to W.2.
>
> >**Q.3: Context vector dimension & pooling, structure of the gating MLP**
>
>
> We thank the reviewer for this insightful question regarding the hyper-parameter settings and structural design choices of our model. In response, we have conducted comprehensive ablation studies to verify the impact of the context vector dimensionality, pooling strategies, and the gating MLP structure on model performance.
>
> **1. Dimensionality of the Context Vector (`ctx_dim`)**
> We investigated the sensitivity of the model to the Context Vector Dimension (`ctx_dim`), a critical hyper-parameter in our context encoder. As presented in **Table 6** below, we varied the dimension from 64 to 512.
> The results indicate that performance initially improves as the dimension increases, peaking at **128** with an accuracy of **96.17%**. Increasing the dimension further (to 256 or 512) results in a performance decline, likely attributable to feature redundancy or overfitting given the available training data. Consequently, we identified **128** as the optimal setting, striking a balance between representation capacity and model complexity.
>
> **Table 6: Ablation study on Context Vector Dimension (`ctx_dim`).**
> | `ctx_dim` | 64 | 96 | **128 (Default)** | 256 | 512 |
> | :--- | :---: | :---: | :---: | :---: | :---: |
> | **Accuracy (%)**| 94.07 | 95.12 | **96.17** | 94.07 | 93.73 |
>
> **2. Pooling Strategies in DSTA4d**
> Regarding the aggregation method used in the DSTA4d module, we compared our proposed strategy against standard global pooling operations (Mean Pooling and Max Pooling). As shown in **Table 7**, our method outperforms Max Pooling by **0.7%** and Mean Pooling by **1.75%**. This demonstrates that our specific design captures the spatiotemporal context more effectively than simple statistical aggregation.
>
> **Table 7: Comparison of Pooling Strategies.**
> | Strategy | Mean Pooling | Max Pooling | **Ours** |
> | :--- | :---: | :---: | :---: |
> | **Accuracy (%)**| 94.42 | 95.47 | **96.17** |

---

### Official Review · Reviewer_QQPt · 2025-10-28

**Soundness:** 3
**Presentation:** 2
**Contribution:** 2
**Rating:** 4
**Confidence:** 4

**Summary:**

The paper proposes a dynamic spatio-temporal decoupling model (DSTA4D) for 4D point cloud video understanding. It separates spatial and temporal features at the embedding layer and employs a Dynamic Spatio-Temporal Adapter (DST-Adapter) to achieve content-adaptive fusion through context gating and FiLM modulation.

**Strengths:**

1.	This paper clearly identifies a real limitation of existing 4D models, i.e., the static and uniform computation graphs, and argues for the need of content-aware computation allocation. The idea sounds appealing.

2.	The modular design (embedding decoupling + DST-Adapter + FiLM) is easy to understand and implement.

**Weaknesses:**

1.	The model claims to adapt computation based on input content, using a global context vector (ctx) that generates gating weights through an MLP and fuses them via FiLM modulation. However, there’s no actual analysis or visualization showing how these gates behave across different samples or datasets. We don’t see whether the gating weights really change in a meaningful way, or how this adaptivity connects to the reported accuracy gains.

2.	Although the paper emphasizes “intelligent computation allocation” and “adaptive efficiency,” it does not report FLOPs, memory consumption, inference latency, or throughput.

3.	The proposed “decoupled spatio-temporal embedding” is presented as a key contribution, but similar ideas have been widely used in previous 4D models. .Besides, there is no ablation comparing the decoupled and non-decoupled variants to justify its impact in this paper.

4.	The ablation only removes FiLM and the Adapter. It does not analyze the relative importance or complementarity of the three branches (identity/spatial/temporal), nor test the sensitivity to context statistics (mean/max pooling) or gating parameters.

5.	Some figures, equations, and tables are not well standardized.

**Questions:**

1. How do the gating distributions differ across categories, scenes, or datasets? Can this be visualized?

2. What are the FLOPs, parameter counts, memory usage, inference latency, and throughput?

3. What differentiates this decoupling scheme from similar 4D spatio-temporal embeddings in prior work?

4. Have you conducted an ablation comparing the coupled vs. decoupled variants to quantify its contribution?

---

> ### Author Response · Authors · 2025-11-21
> **Response to Reviewer QQPt (Part 1)**
>
> > **W.1: Lack of gating analysis and visualization.**
>
> We sincerely thank the reviewer for raising this important point. We agree that exploring the internal dynamics of the gating mechanism is crucial for validating our claims of adaptivity. The proposed DST-Adapter utilizes a global context vector (`ctx`) combined with FiLM modulation to dynamically fuse features across streams. This design allows the model to assign importance weights ($\alpha$: Identity, $\beta$: Spatial, $\gamma$: Temporal) adaptively based on input complexity.
>
> Following your suggestion, we conducted a statistical analysis of the gating weights' evolution during the training process. As detailed in **Tables 1 and 2**, the gating mechanism exhibits distinct evolutionary trajectories across different datasets, confirming its "content-aware" capability:
>
> 1.  **Stable Spatial Dominance on MSR-Action3D (Table 1):**
>     On the MSR dataset, the model demonstrates a stable convergence pattern. The spatial weight ($\beta$) steadily increases from an initial 0.347 to 0.525, while the temporal weight ($\gamma$) and identity weight ($\alpha$) gradually decrease. This indicates that for the MSR dataset, the model identifies spatial geometry as the primary discriminative cue early in training and maintains this focus, requiring less aggressive feature transformation.
>
> 2.  **Dynamic Temporal Adaptation on NTU RGB+D (Table 2):**
>     Conversely, the NTU dataset elicits a more complex adaptation strategy. We observe a distinct **"temporal surge"** in the mid-training phase (Epochs 5–8), where the temporal weight ($\gamma$) peaks at roughly 0.50. This suggests the model prioritizes learning complex temporal dynamics during the critical learning period. Subsequently, the focus shifts towards refining spatial details ($\beta \rightarrow 0.567$), while the identity weight drops significantly ($\alpha < 0.08$). This low $\alpha$ indicates that the complex nature of NTU requires deep feature transformation rather than preserving raw input features.
>
>
> These comparative results demonstrate that the gating weights are not static parameters but evolve stochastically in response to the learning progress and data characteristics. The mechanism effectively acts as a dynamic selector—emphasizing temporal branches for high-motion complexity and spatial branches for geometric-heavy samples. We have included these quantitative results and the corresponding visualizations in **Appendix Fig. 7** of the revised manuscript (highlighted in red).
>
> **Table 1: Evolution of Gating Weights on MSR-Action3D**
>
> | Epoch | $\alpha$ (Identity) | $\beta$ (Spatial) | $\gamma$ (Temporal) |
> | :---: | :---: | :---: | :---: |
> | **1** | 0.325 | 0.347 | 0.328 |
> | **5** | 0.299 | 0.455 | 0.247 |
> | **10** | 0.291 | 0.489 | 0.220 |
> | **15** | 0.282 | 0.508 | 0.210 |
> | **30** | 0.267 | 0.525 | 0.207 |
>
> **Table 2: Evolution of Gating Weights on NTU RGB+D**
>
> | Epoch | $\alpha$ (Identity) | $\beta$ (Spatial) | $\gamma$ (Temporal) |
> | :---: | :---: | :---: | :---: |
> | **1** | 0.394 | 0.338 | 0.268 |
> | **5** | 0.213 | 0.326 | 0.461 |
> | **8** | 0.134 | 0.365 | 0.501 |
> | **10** | 0.113 | 0.442 | 0.445 |
> | **13** | 0.080 | 0.567 | 0.353 |

---

> ### Author Response · Authors · 2025-11-21
> **Response to Reviewer QQPt (Part 2)**
>
> > **W.2: No efficiency metrics.**
>
> We sincerely thank the reviewer for raising this important concern. You noted that although the paper emphasizes “intelligent computation allocation” and “adaptive efficiency,” it does not report FLOPs, memory consumption, inference latency, or throughput. To address this gap, we conducted additional experiments under a unified hardware environment, comparing PSTTransformer and our proposed DSTA4D, with each metric measured three times to ensure statistical stability.
>
> The results in Table 3 show that DSTA4D incurs modest increases in parameters (+5.4%) and FLOPs (+3.1%), which stem from the introduction of the dynamic spatio-temporal adaptation module. However, in terms of inference efficiency, DSTA4D performs better: it reduces latency by about 3.2%, improves throughput by 3.5%, and correspondingly increases frame rate. These findings indicate that the adaptive module enhances representational capacity without introducing noticeable computational overhead, and even achieves slight efficiency gains in practice.
>
> Most notably, DSTA4D reduces GPU cached memory usage to 392 MB, representing a 28.7% reduction compared to PSTTransformer. This highlights the superior memory efficiency of DSTA4D, which makes it particularly suitable for scenarios involving multi-model parallelism or deployment under memory-constrained environments.
>
> In summary, although DSTA4D introduces a small increase in computational cost, it demonstrates clear advantages in latency, throughput, and memory efficiency. These results provide strong evidence that the proposed dynamic spatio-temporal adaptation module achieves a well-balanced trade-off between adaptability and practical efficiency.
>
>
> **Table 3. Efficiency Comparison between PSTTransformer and DSTA4D (3 runs, mean ± std, relative change vs. PST)**
>
> | Model                  | Params (M) ▼ | FLOPs (G) ▼ |     Latency (ms) ▼| Throughput (clips/s) ▲ |      Frames (fps) ▲ | GPU alloc (MB) ▼| GPU cached (MB) ▼|
> | ------------------ | ----------:| ---------:| ------------:| --------------------:| ------------:| --------------:| ---------------:|
> | **PSTTransformer** | 45.483     | 40.790    | 33.28 ± 0.14 | 30.05 ± 0.14         | 721.19 ± 3.0 | 182.22         | 550             |
> | **DSTA4D (Ours)**  | 47.942 (+5.40%) | 42.054 (+3.10%) | **32.21 ± 1.65 (−3.22%)** | **31.10 ± 1.45 (+3.50%)** | **746.45 ± 34.8 (+3.50%)** | 191.83 (+5.27%) | **392 (−28.73%)** |

---

> ### Author Response · Authors · 2025-11-21
> **Response to Reviewer QQPt (Part 3)**
>
> > **W.3: Decoupling contribution unclear / no ablation.**
>
> We sincerely thank the reviewer for this valuable comment. We fully understand your concern regarding the similarity of the proposed “decoupled spatio-temporal embedding” to prior work and the lack of ablation studies. While the concept of spatio-temporal decoupling has been explored in earlier works such as PSTNet, our proposed Decoupled Spatio-Temporal Embedding differs fundamentally in both mechanism and purpose. Prior approaches typically perform static separation at deeper layers of the network, whereas we adopt a pre-embedding strategy, aiming to construct a spatio-temporal feature space with approximate error orthogonality at the input stage (see Appendix A.1). This early-stage decoupling helps suppress cross-error propagation. More importantly, our decoupling serves as a prerequisite for the content-aware mechanism, enabling the DST-Adapter to dynamically generate gating weights (𝛼,𝛽,𝛾) and allocate computational resources according to input complexity, rather than relying on a static “one-size-fits-all” computational graph
>
> Regarding ablation studies, Table 4 provides analysis of the Decoupled Spatio-Temporal Adapter as a whole. Results show that removing this module (which includes both decoupled embedding and dynamic fusion) leads to a significant drop in accuracy on MSR-Action3D, from 96.17% to 93.37% (−2.8%). This performance degradation strongly supports the necessity of explicit decoupling combined with dynamic gating for capturing long-term spatio-temporal dependencies.
>
> We further conducted detailed ablations on the three decoupled branches (Identity, Spatial, Temporal) to verify their importance and complementarity. Results demonstrate that the full model (96.17%) consistently outperforms any variant missing a single branch:
> 1. Removing the Spatial branch → accuracy drops to 93.72%, highlighting the importance of spatial geometric interactions.
> 2. Removing the Temporal branch → accuracy drops to 94.42%, confirming the critical role of independent temporal modeling.
> 3. Removing the Identity branch → accuracy drops to 95.47%, showing that retaining the raw feature stream is necessary for information integrity and training stability.
>
> Compared to the baseline normal decoupling (93.37%), our multi-branch dynamic collaboration strategy achieves a clear performance gain. This demonstrates that the branches are not simply additive, but rather complementary, jointly enhancing the model’s ability to represent complex spatio-temporal data.
>
> **Table 4. Ablation Result of decoupling.**
> | Model Setting              | Accuracy (%) |
> | :------------------------- | -----------: |
> | **Our Method (Full Model)**| **96.17**    |
> | w/o Identity branch        | 95.47        |
> | w/o Temporal branch        | 94.42        |
> | w/o Spatial branch         | 93.72        |
> | Normal decoupling (Baseline)| 93.37       |

---

> ### Author Response · Authors · 2025-11-21
> **Response to Reviewer QQPt (Part 4)**
>
> > **W.4: Branches & statistics sensitivity untested.**
>
> We sincerely thank the reviewer for this insightful comment. You pointed out that our ablation only removed FiLM and the Adapter, without analyzing the relative importance and complementarity of the three branches (Identity/Spatial/Temporal), or testing the sensitivity to context statistics (Mean/Max pooling) and gating parameters. To address this concern, we conducted additional experiments to carefully evaluate these aspects.
>
> For context statistics, we compared the default Mean + Max pooling strategy with variants using Mean Pooling only or Max Pooling only. The results are summarized in the table below. In addition, as shown in Table 4, we conducted a structured ablation study on the three decoupled branches (Identity, Spatial, Temporal) to verify their relative importance and complementarity (see Response to W.3). Taken together, these experiments demonstrate that the model is not only sensitive to the choice of context statistics, but also that each decoupled branch plays an indispensable role in boosting overall performance.
>
> **Table 5. Ablation Result of FiLM and the Adapter.**
> | Context Statistics        | Accuracy (%) | Performance Drop |
> | :------------------------ | -----------: | ---------------: |
> | **Mean + Max (Full Model)** | **96.17**    | -                |
> | Max Pooling Only          | 95.47        | -0.70%           |
> | Mean Pooling Only         | 94.42        | -1.75%           |
>
> The observed performance drops (e.g., -1.75\%) confirm the model's sensitivity to context descriptors and highlight the complementary nature of different statistics. Specifically, while mean pooling captures global distributions but tends to smooth out motion details, and max pooling extracts salient features yet misses average information, their combination achieves optimal performance (96.17\%). This synergy effectively integrates the "general state" with "salient features" to inform the dynamic gating mechanism, demonstrating that our design choices regarding branch decoupling and context statistics are both justified and essential for the effectiveness of the proposed approach.
>
>
> > **W.5: Figures/tables/equations non-standardized.**
>
> We sincerely thank the reviewer for the valuable comments regarding the standardization of figures, tables, and equations. Following your suggestions, we carefully reviewed the entire manuscript and unified the formatting of all visual and mathematical elements, correcting inconsistencies to improve readability and clarity.
>
> All these changes have been implemented in the revised manuscript, with key updates highlighted. We believe these refinements will improve the overall presentation of the paper and facilitate better understanding for readers.
>
>
>
> > **Q.1: Lack of gating analysis and visualization.**
>
> see W.1.
>
> > **Q.2: No efficiency metrics.**
>
> see W.2.
>
> > **Q.3: Decoupling contribution unclear / no ablation.**
>
> see W.3.
>
> > **Q.4: Branches & statistics sensitivity untested.**
>
> see W.4.

---

### Official Review · Reviewer_L1Fq · 2025-10-29

**Soundness:** 3
**Presentation:** 3
**Contribution:** 3
**Rating:** 6
**Confidence:** 4

**Summary:**

This paper proposes DSTA4D, a content-aware framework for 4D point cloud video understanding designed to address the inefficiencies of static, monolithic models. The core contribution is the Dynamic Spatio-Temporal Adapter (DST-Adapter), a lightweight module that dynamically computes a global context vector from the input sequence. This vector is then used to generate gating weights that adaptively fuse features from three parallel streams: an identity path, a spatial enhancement path, and a temporal enhancement path. Experiments across multiply benchmarks demonstrate its effectiveness.

**Strengths:**

1.	This paper presents a novel and well-motivated framework, DSTA4D, which tackles the inefficiency of static 4D point cloud models. This content-aware mechanism well addresses the one-size-fits-all limitation of prior work.
2.	Extensive experiments validate the effectiveness of the proposed method.
3.	This paper is written and organized well.

**Weaknesses:**

1.	The paper claims to solve inefficient resource allocation, but provides no efficiency metrics (FLOPs, latency, parameters). It will be more convincing to provide some efficiency metrics.
2.	The ST Transformer backbone is not defined in the main paper. Without knowing its architecture (details are hidden in Appendix A.2), it's impossible to discern how much performance gain comes from the novel adapter versus a potentially strong, custom backbone.
3.	The DST-Adapter is proposed as a general module but is only tested on its own Transformer backbone. It will be more interesting to plug it into other architectures.

**Questions:**

Refer to the Weakness.

---

> ### Author Response · Authors · 2025-11-21
> **Response to Reviewer L1Fq (Part 1)**
>
> >**W.1: No efficiency metrics.**
>
> We sincerely thank the reviewer for raising this important concern. You noted that although the paper emphasizes “intelligent computation allocation” and “adaptive efficiency,” it does not report FLOPs, memory consumption, inference latency, or throughput. To address this gap, we conducted additional experiments under a unified hardware environment, comparing PSTTransformer and our proposed DSTA4D, with each metric measured three times to ensure statistical stability.
>
> The results in Table 1 show that DSTA4D incurs modest increases in parameters (+5.4%) and FLOPs (+3.1%), which stem from the introduction of the dynamic spatio-temporal adaptation module. However, in terms of inference efficiency, DSTA4D performs better: it reduces latency by about 3.2%, improves throughput by 3.5%, and correspondingly increases frame rate. These findings indicate that the adaptive module enhances representational capacity without introducing noticeable computational overhead, and even achieves slight efficiency gains in practice.
>
> Most notably, DSTA4D reduces GPU cached memory usage to 392 MB, representing a 28.7% reduction compared to PSTTransformer. This highlights the superior memory efficiency of DSTA4D, which makes it particularly suitable for scenarios involving multi-model parallelism or deployment under memory-constrained environments.
>
> In summary, although DSTA4D introduces a small increase in computational cost, it demonstrates clear advantages in latency, throughput, and memory efficiency. These results provide strong evidence that the proposed dynamic spatio-temporal adaptation module achieves a well-balanced trade-off between adaptability and practical efficiency. We have included these detailed efficiency measurements in the revised manuscript to comprehensively support our claims on “adaptive efficiency.”
>
>
> **Table 1. Efficiency Comparison between PSTTransformer and DSTA4D (3 runs, mean ± std, relative change vs. PST)**
>
> | Model                  | Params (M) ▼ | FLOPs (G) ▼ |     Latency (ms) ▼| Throughput (clips/s) ▲ |      Frames (fps) ▲ | GPU alloc (MB) ▼| GPU cached (MB) ▼|
> | ------------------ | ----------:| ---------:| ------------:| --------------------:| ------------:| --------------:| ---------------:|
> | **PSTTransformer** | 45.483     | 40.790    | 33.28 ± 0.14 | 30.05 ± 0.14         | 721.19 ± 3.0 | 182.22         | 550             |
> | **DSTA4D (Ours)**  | 47.942 (+5.40%) | 42.054 (+3.10%) | **32.21 ± 1.65 (−3.22%)** | **31.10 ± 1.45 (+3.50%)** | **746.45 ± 34.8 (+3.50%)** | 191.83 (+5.27%) | **392 (−28.73%)** |
>
> >**W.2: Unclear contribution.**
>
> We sincerely thank the reviewer for the professional insights. We wish to clarify that the ST Transformer utilizes the PSTTransformer architecture as its backbone. To enhance the clarity of the overall framework, we have provided a detailed decomposition of the ST Transformer’s backbone structure in the Appendix. This backbone adheres to standard spatio-temporal Transformer designs; the detailed breakdown in the Appendix is intended solely to facilitate a more intuitive understanding of the functional role of each module within the overall workflow, rather than to introduce any novel network architectures.
>
> Specifically, the point cloud sequence first passes through a Spatio-Temporal Embedding module, where local geometric and temporal features are extracted via spatial and temporal convolutions. The extracted features are then fed into the DST-Adapter, which comprises a SpatialAdapter (incorporating token mixing and channel-wise LoRA) and a TemporalAdapter (utilizing depthwise temporal convolution). Simultaneously, a lightweight ContextEncoder captures global statistics, generates dynamic gating weights ($\alpha, \beta, \gamma$) via SiLU activation, and performs adaptive feature scaling and shifting through FiLM modulation. Subsequently, the enhanced features are processed by the ST Transformer layers, each consisting of standard Attention and FeedForward blocks.We emphasize that the structural details in the Appendix are strictly for visualization and ease of understanding. Furthermore, our ablation studies confirm that all observed performance improvements stem from the proposed adapter, rather than the backbone network itself.

---

> ### Author Response · Authors · 2025-11-21
> **Response to Reviewer L1Fq (Part 2)**
>
> >**W.3: No cross-architecture test.**
>
> We sincerely thank the reviewer for this insightful suggestion regarding the universality of our design. To verify the "plug-and-play" capability and generalization of our proposed GSTA4d (DSTA) module, we integrated it into two representative point cloud video modeling backbones: **P4Transformer** and **PST-Transformer**. We conducted ablation studies to measure the performance gains on these baselines.
>
> As shown in **Table 2**, our module consistently improves the performance of different architectures:
> * **On P4Transformer:** The integration of our module yields an accuracy of **93.03%**, achieving a significant improvement of **+2.09%** over the baseline (90.94%).
> * **On PST-Transformer:** Our method boosts the accuracy to **96.17%**, resulting in a **+2.44%** performance gain compared to the original model (93.73%).
>
> **Table 2: Plug-and-play performance comparison on P4Transformer and PST-Transformer backbones.**
>
> | Method | Accuracy (%) |
> | :--- | :--- |
> | P4Transformer | 90.94 |
> | P4Transformer + DSTA | **93.03** |
> | PST-Transformer | 93.73 |
> | PST-Transformer + DSTA | **96.17** |
>
>
> These results empirically demonstrate that our adaptive decoupling design is robust and effective across different backbones, supporting its potential as a universal module for point cloud video understanding.
>
>
> >**Questions**
>
> see Weekness.

---

### Author Response · Authors · 2025-12-03
**Summary of Revisions**

We sincerely thank the reviewers for their constructive feedback. During the rebuttal phase, we have conducted extensive additional experiments and revisions to address all concerns regarding efficiency, generalization, mechanism interpretability, and novelty. All revisions are included in the revised paper highlighted in red. The key updates are summarized below:

**1. Comprehensive Efficiency Analysis**

We addressed the common concern regarding computational overhead by providing detailed metrics (FLOPs, Latency, Throughput, Memory).
* **Result:** While DSTA4D has a slight increase in parameters (+5.4%), it achieves **lower inference latency (-3.2%)** and **higher throughput (+3.5%)** compared to the baseline.
* **Key Advantage:** Most notably, it reduces **GPU cached memory usage by 28.7%** (392MB vs. 550MB), validating our claim of "adaptive efficiency" and suitability for memory-constrained environments.

**2. New Benchmarks and SOTA Performance**

To demonstrate robustness on complex, diverse datasets, we extended our evaluation to the **HOI4D** dataset (large-scale human-object interaction).
* **Performance:** Our method achieves **78.8%** accuracy on the standard setting. Furthermore, under the multi-modal training setting (utilizing RGB priors), we achieved **85.4%**, surpassing state-of-the-art methods like X4D-SceneFormer (84.1%) and PPTr+C2P(81.1%).

**3. Validation of "Plug-and-Play" Generalization**

We verified the universality of the DSTA module by integrating it into different backbones:
* **P4Transformer:** Accuracy improved from 90.94% to **93.03% (+2.09%)**.
* **PST-Transformer:** Accuracy improved from 93.73% to **96.17% (+2.44%)**.
This confirms that our module provides consistent gains across different architectures.

**4. In-depth Analysis of the Dynamic Gating Mechanism**

We provided empirical evidence of the "intelligent allocation" mechanism by tracking gating weights ($\alpha, \beta, \gamma$) during training.
* **Observation:** The model exhibits distinct behaviors: "Spatial Dominance" on MSR-Action3D vs. a "Temporal Surge" on NTU RGB+D. This proves the model dynamically prioritizes spatial or temporal branches based on data complexity, rather than using a static graph.

**5. Clarifications on Novelty and Design**

* **Systemic Structural Innovation:** We clarified that DSTA4D is not a mere combination of modules, but a novel **"context-driven three-path dynamic decoupling" paradigm**. This design fundamentally addresses the limitations of rigid "two-stream" decoupling strategies, which often introduce noise in static scenes or fail to capture sufficient features in high-dynamic ones.
* **Intelligent Computational Allocation:** We emphasized that the core novelty lies in the **sample-level intelligent computational allocation**. By synergizing context statistics, gating, and FiLM, our method dynamically shifts modeling focus (e.g., activating temporal paths only for high-motion scenes), ensuring superior adaptability compared to static backbones.
* **Ablations:** Extensive ablations on kernel sizes ($k=3$ is optimal), pooling strategies, and branch removal confirm the necessity of each component.

We believe these revisions and additional results robustly support the effectiveness and novelty of DSTA4D.

---

### Meta-Review · Area_Chair_uuMD · 2025-12-18

**Summary:**

The recommendation for this paper is primarily informed by concerns regarding the experimental completeness and the depth of technical innovation. While the reviewers appreciated the strong empirical results on MSR-Action3D and the authors provided efficiency metrics and gating visualizations during the rebuttal, several critical issues remain:

Missing Baselines and Paradigms: From the meta reviewer’s perspective, the paper fails to benchmark against the direct 4D modeling paradigm (e.g., [1] [2] [3]). Without comparing the proposed "adaptive decoupling" strategy against unified 4D convolution approaches, the fundamental motivation of the framework remains unproven.

[1] A Unified Framework for Human-centric Point Cloud Video Understanding. CVPR 2024

[2] Dynamic 3D point cloud sequences as 2D videos. TPAMI 2024

[3] Uni4D: Unifying Visual Foundation Models for 4D Modeling from a Single Video. CVPR 2025

**Limited Technical Novelty Beyond Existing Components.**

Reviewers noted that the core technical design primarily integrates established mechanisms—such as FiLM-style feature modulation, gating-based fusion, and lightweight adapter modules—within a spatio-temporal transformer backbone. Although the integration is coherent and effective, the reviewers questioned whether this constitutes a substantial conceptual advance over prior adaptive or modular spatio-temporal architectures, rather than an incremental recombination of known techniques.

**Restricted Evaluation Scope and Generalization Evidence.**

As pointed out by Reviewer GSr5, the evaluation focuses on human action recognition datasets and a synthetic driving dataset (Synthia4D). The absence of experiments on more complex, large-scale real-world outdoor scenarios (e.g., dynamic LiDAR benchmarks) limits the evidence for the method’s generality and robustness in broader 4D perception settings.

**Unclear Practical Impact of the Adaptive Mechanism.**

Although the rebuttal reports modest improvements in latency and memory usage, reviewers expressed reservations about whether the proposed content-aware allocation provides a practically significant advantage over well-optimized static models. In particular, since the dynamic gating mechanism primarily reweights feature branches rather than explicitly skipping computation, the real-world efficiency benefits of the added architectural complexity remain uncertain.

**Reviewer Concerns:**

The rebuttal addressed efficiency metrics and gating analysis, but concerns about novelty, paradigm-level comparison, evaluation scope, and practical impact remain.

**Reviewer Scores:**

Reviewer cGvm: 4 — Finds the contribution insufficiently novel and poorly positioned among 4D modeling approaches.

Reviewer GSr5: 4 — Evaluation remains too narrow to support strong generalization claims.

Reviewer QQPt: 4 — Rebuttal addresses details but does not alter concerns about incremental novelty.

Reviewer L1Fq: 6 — Maintains a weakly positive view that does not outweigh broader concerns.

---

### Decision · Program_Chairs · 2026-01-26

Reject